# EXPECTED TIGHT BOUNDS FOR ROBUST DEEP NEURAL NETWORK TRAINING

## ABSTRACT

Training Deep Neural Networks (DNNs) that are robust to norm bounded adversarial attacks remains an elusive problem. While verification based methods are generally too expensive to robustly train large networks, it was demonstrated by Gowal et al. (2019) that bounded input intervals can be inexpensively propagated from layer to layer through deep networks. This interval bound propagation (IBP) approach led to high robustness and was the first to be employed on large networks. However, due to the very loose nature of the IBP bounds, particularly for large/deep networks, the required training procedure is complex and involved. In this paper, we closely examine the bounds of a block of layers composed of an affine layer, followed by a ReLU, followed by another affine layer. To this end, we propose *expected* bounds (true bounds in expectation), which are provably tighter than IBP bounds in expectation. We then extend this result to deeper networks through blockwise propagation and show that we can achieve orders of magnitudes tighter bounds compared to IBP. Using these tight bounds, we demonstrate that a simple standard training procedure can achieve impressive robustness-accuracy trade-off across several architectures on both MNIST and CIFAR10.

## 1 INTRODUCTION

Deep neural networks (DNNs) have demonstrated impressive performance in many fields of research with applications ranging from image classification (Krizhevsky et al., 2012; He et al., 2016) and semantic segmentation (Long et al., 2015) to speech recognition (Hinton et al., 2012), just to name a few. Despite this success, DNNs are still susceptible to small imperceptible perturbations, which can lead to drastic performance degradation, especially in visual classification tasks. Such perturbations are best known and commonly referred to as *adversarial attacks*. Early work showed that simple algorithms (*e.g.* maximizing the classification loss with respect to the input using a single optimization iteration (Goodfellow et al., 2014)) can easily construct such adversaries. Since then, a research surge has emerged to develop simple routines to construct adversarial examples consistently. For instance, Moosavi-Dezfooli et al. (2016) proposed a simple algorithm, called DeepFool, which finds the smallest perturbation that fools a linearized version of the network. Interestingly, the work of Moosavi-Dezfooli et al. (2017) demonstrated that such adversaries can be both network and input agnostic, *i.e.* universal deterministic samples that fool a wide range of DNNs across a large number of input samples. More recently, it was shown that such adversaries can also be as simple as Gaussian noise (Bibi et al., 2018). Knowing that DNNs are easily susceptible to simple attacks can hinder the public confidence in them, especially for real-world deployment, *e.g.* in self-driving cars and devices for the visually impaired.

Such a performance nuisance has prompted several active research directions, in particular, work towards network defense and verification. Network defense aims to train networks that are robust against adversarial attacks through means of robust training or procedures at inference time that dampen the effectiveness of the attack (Madry et al., 2018; Wong & Kolter, 2018; Raghunathan et al., 2018; Alfadly et al., 2019). On the other hand, verification aims to certify/verify for a given DNN that there exists no small perturbation of a given input that can change its output prediction (Katz et al., 2017; Sankaranarayanan et al., 2016; Weng et al., 2018a). However, there are also works at the intersection of both often referred to as robustness verification methods, which use verification methods to train robust networks. Such algorithms often try to minimize the exact (or upper bound) of the worst adversarial loss over all possible bounded energy (often measured in $\ell_\infty$ norm) perturbation around a given input.

Although verification methods prove to be effective in training robust networks (Wong & Kolter, 2018), they are computationally expensive, thus limiting their applicability to only small, at best medium, sized networks. However, Gowal et al. (2019) recently demonstrated that robustly training large networks is possible by leveraging the cheap-to-compute but very loose interval-based verifier, known as interval domain from Mirman et al. (2018). In particular, they propagate the $\epsilon$-$\ell_\infty$ norm bounded input centered at $\mathbf{x} \in \mathbb{R}^n$, *i.e.* $[\mathbf{x} - \epsilon\mathbf{1}_n, \mathbf{x} + \epsilon\mathbf{1}_n]$, through every layer in the network at a time. This interval bound propagation (IBP) is inexpensive and simple; however, it results in very loose output interval bounds, which in turn necessitates a complex and involved training procedure.

Closer to our work, there has been several prior arts that propose to perform verification differently. In particular, Webb et al. (2018) presents a statistical approach to assessing robustness of neural networks. As opposed to verification methods, which in many cases can be hard to scale, that provide a binary measure of robustness per sample, they propose to frame verification as a probability of violation instead. That is to say, they investigate the probability of failure over a violation rather than confirming that this probability is exactly zero. Moreover, (Weng et al., 2018b) propose CLEVER which estimates a lower bound to the minimum perturbation rather that finding the lower bounds exactly. In both works, the verification is tackled by, and closely related to our direction of work, probabilistically estimating the bounds.

In this paper, we are interested in improving the tightness of output interval bounds (referred to as bounds from now on). We do so by closely examining the bounds for a block of layers composed of an affine layer, followed by a ReLU nonlinearity, followed by another affine layer under $\epsilon$-$\ell_\infty$ bounded input. In fact, we propose new *expected* bounds for this block of layers, which we prove to be not only supersets to the true bounds of this block in expectation but also very tight to the true bounds. Lastly, we show how to extend such a result to deeper networks through blockwise bound propagation leading to several orders of magnitude tighter bounds as compared to IBP.

**Contributions.** Our contributions are three-fold. (**i**) We propose new bounds for the block of layers composed of an affine layer, followed by a ReLU, followed by another affine layer. We prove that these bounds are in expectation, under a distribution of network parameters, supersets to the true bounds of this block. Moreover, we prove that these bounds are much tighter, in expectation (will be formalized later) than the IBP bounds (Gowal et al., 2019) generated by propagating the input bounds through every layer in the block. Our bounds get even tighter as the number of hidden nodes in the first affine layer increases. (**ii**) We show a practical and efficient approach to propagate our bounds (for the block of layers) through blocks (not through individual layers) of a deep network, thus resulting in magnitudes tighter output bounds compared to IBP. (**iii**) We conduct experiments on synthetic networks and on real networks, to verify the theory, as well as the factors of improvement over IBP. Due to the tightness of our proposed expected bounds, we show that with a simple standard training procedure, large/deep networks can be robustly trained on both MNIST (LeCun, 1998) and CIFAR10 (Krizhevsky & Hinton, 2009) achieving state-of-art robustness-accuracy trade-off compared to IBP. In other words, we can consistently improve robustness by significant margins with minimal effect on test accuracy as compared to IBP.

## 2 RELATED WORK

Training accurate and robust DNNs remains an elusive problem, since several works have demonstrated that small imperceptible perturbations (adversarial attacks) to the DNN input can drastically affect their performance. Early works showed that with a very simple algorithm, as simple as maximizing the loss with respect to the input for a single iteration (Goodfellow et al., 2014), one can easily construct such adversaries. This has strengthened the line of work towards network verification for both evaluating network robustness and for robust network training. In general, verification approaches can be coarsely categorized as exact or relaxed verifiers.

**Exact Verification.** Verifiers of this type try to find the exact largest adversarial loss over all possible bounded energy (usually measured in $\ell_\infty$ norm) perturbations around a given input. They are often tailored for piecewise linear networks, *e.g.* networks with ReLU and LeakyReLU nonlinearities. They typically require mixed integer solvers (Cheng et al., 2017; Lomuscio & Maganti, 2017; Tjeng & Tedrake, 2019) or Satisfiability Modulo Theory (SMT) solvers (Xiaowei Huang & Wu, 2017; Ehlers, 2017). The main advantage of these approaches is that they can reason about exact adversarial robustness; however, they generally are computationally intractable for verification purposes let alone any sort of robust network training. The largest network used for verification with such verifiers was with the work of Tjeng & Tedrake (2019), which employed a mixed integer solver applied to

networks of at most 3 hidden layers. The verification is fast for networks that are pretrained with a relaxed verifier but gets much slower on normally trained similar sized networks.

**Relaxed Verification.** Verifiers of this type aim to find an upper bound on the worst adversarial loss across a range of bounded inputs. For instance, a general framework called CROWN was proposed by Huan Zhang & Daniel (2018) to certify robustness by bounding the activation with linear and quadratic functions, thus, enabling the study of generic, not necessarily piecewise linear, activation functions. By utilizing the structure in ReLU based networks, the work of Weng et al. (2018a) proposed two fast algorithms based on linear approximation on the ReLU units. Moreover, Wang et al. (2018c) proposed ReluVal for network verification based on symbolic interval bounds, while Wang et al. (2018b) proposed Neurify with much tighter bounds. Several other works utilized the dual view of the verification problem (Wong & Kolter, 2018; Wong et al., 2018). More recently, Salman et al. (2019) unified a large number of recent works in a single convex relaxation framework and revealed several relationships between them. In particular, it was shown that convex relaxation methods that fit this framework suffer from an inherent barrier compared to exact verifiers.

For completeness, it is important to note that there are also hybrid methods that combine both exact and relaxed verifiers and have shown to be effective (Rudy Bunel, 2018). Although relaxed verifiers are much more computationally friendly than exact verifiers, they are still too expensive for robust training of large/deep networks (with more than 5 hidden layers). However, very loose relaxed verifiers can possibly still be exploited for this purpose. For instance, Wang et al. (2018a) leveraged symbolic interval analysis to verifiably train large networks. More recently, the work of Gowal et al. (2019) proposed to use an inexpensive but very loose interval bound propagation (IBP) certificate to train (for the first time) large robust networks with state-of-the-art robustness performance. This was at the expense of a complex and involved training routine resulting from the loose nature of the bounds. To remedy these training difficulties, we instead propose *expected* bounds, not for each layer individually, but for a block of layers jointly. Such bounds are slightly more expensive to compute but are much tighter in expectation. We then propagate these bounds through every block in a deeper network to attain much tighter bounds overall as compared to layerwise IBP. The tighter bounds enable the use of simple standard training routines for robust training of large networks, resulting in state-of-art robustness-accuracy trade-off.

## 3 Expected Tight Interval Bounds

We analyze the interval bounds of a DNN by proposing expected true and tight bounds for a two-layer network (Affine-ReLU-Affine). Then, we propose a mechanism to extend them for deeper networks. First, we detail the interval bounds of Gowal et al. (2019) to put our bounds in context.

### 3.1 Interval Bounds for a Single Affine Layer

For a single affine layer parameterized by $\mathbf{A}_1 \in \mathbb{R}^{k \times n}$ and $\mathbf{b}_1 \in \mathbb{R}^k$, it is easy to show that its output lower and upper interval bounds for an $\epsilon$-$\ell_\infty$ norm bounded input $\tilde{\mathbf{x}} \in [\mathbf{x} - \epsilon \mathbf{1}_n, \mathbf{x} + \epsilon \mathbf{1}_n]$ are:

$$\mathbf{l}_1 = \mathbf{A}_1 \mathbf{x} + \mathbf{b}_1 - \epsilon |\mathbf{A}_1| \mathbf{1}_n, \quad \mathbf{u}_1 = \mathbf{A}_1 \mathbf{x} + \mathbf{b}_1 + \epsilon |\mathbf{A}_1| \mathbf{1}_n. \tag{1}$$

Note that $|.|$ is an elementwise absolute operator. In the presence of any non-decreasing elementwise nonlinearity (*e.g.* ReLU), the bounds can then be propagated by applying the nonlinearity to $\{\mathbf{l}_1, \mathbf{u}_1\}$ directly. As such, the interval bounds can be propagated through the network one layer at a time, as proposed by Gowal et al. (2019). While this interval bound propagation (IBP) mechanism is a very simple and inexpensive approach to compute bounds, these bounds can be extremely loose for deep networks, requiring a complex and involved robust network training procedure.

### 3.2 Proposed Interval Bounds for an Affine-ReLU-Affine Block

Here, we consider a block of layers of the form Affine-ReLU-Affine in the presence of $\ell_\infty$ perturbations at the input. The functional form of this network is: $g(\mathbf{x}) = \mathbf{a}_2^\top \max (\mathbf{A}_1 \mathbf{x} + \mathbf{b}_1, \mathbf{0}) + b_2$, where $\max(.)$ is an elementwise operator. The affine mappings can be of any size, and throughout the paper, we take $\mathbf{A}_1 \in \mathbb{R}^{k \times n}$ and without loss of generality the second affine map is a single vector $\mathbf{a}_2 \in \mathbb{R}^k$. Note that $g$ also includes convolutional layers, since they are also affine mappings.

**Layerwise Interval Bound Propagation (IBP) on $g$.** Here, we apply the layerwise propagation strategy of Gowal et al. (2019) detailed in Section 3.1 on function $g(\tilde{\mathbf{x}})$ with $\tilde{\mathbf{x}} \in [\mathbf{x} - \epsilon \mathbf{1}_n, \mathbf{x} + \epsilon \mathbf{1}_n]$

to obtain bounds $[\mathbf{L_{IBP}}, \mathbf{U_{IBP}}]$. We use these bounds for comparison in what follows.

$$\mathbf{L_{IBP}} = \mathbf{a}_2^\top \left( \frac{\max(\mathbf{u}_1, \mathbf{0}_k) + \max(\mathbf{l}_1, \mathbf{0}_k)}{2} \right) - |\mathbf{a}_2^\top| \left( \frac{\max(\mathbf{u}_1, \mathbf{0}_k) - \max(\mathbf{l}_1, \mathbf{0}_k)}{2} \right) + b_2,$$

$$\mathbf{U_{IBP}} = \mathbf{a}_2^\top \left( \frac{\max(\mathbf{u}_1, \mathbf{0}_k) + \max(\mathbf{l}_1, \mathbf{0}_k)}{2} \right) + |\mathbf{a}_2^\top| \left( \frac{\max(\mathbf{u}_1, \mathbf{0}_k) - \max(\mathbf{l}_1, \mathbf{0}_k)}{2} \right) + b_2.$$

Note that $\max(\mathbf{l}_1, \mathbf{0}_k)$ and $\max(\mathbf{u}_1, \mathbf{0}_k)$ are the result of propagating $[\mathbf{x} - \epsilon\mathbf{1}_n, \mathbf{x} + \epsilon\mathbf{1}_n]$ through the first affine map $(\mathbf{A}_1, \mathbf{b}_1)$ and then through the ReLU nonlinearity, as stated in (1).

**Expected Tight Interval Bounds on $g$.** Our goal is to propose new interval bounds for $g$, as a block, which are tighter than the IBP bounds $[\mathbf{L_{IBP}}, \mathbf{U_{IBP}}]$, since we believe that tighter bounds for a two-layer block, when propagated/extended to deeper networks, can be tighter than applying IBP layerwise. Denoting the true output interval bounds of $g$ as $[\mathbf{L}_{\text{true}}, \mathbf{U}_{\text{true}}]$, the following inequality holds $\mathbf{L}_{\text{true}} \leq g(\tilde{\mathbf{x}}) \leq \mathbf{U}_{\text{true}} \ \forall \tilde{\mathbf{x}} \in [\mathbf{x} - \epsilon\mathbf{1}_n, \mathbf{x} + \epsilon\mathbf{1}_n]$. Deriving these true (and tight) bounds for $g$ in closed form is either hard or results in bounds that are generally very difficult to compute, deeming them impractical for applications such as robust network training. Instead, we propose new closed form expressions for the interval bounds denoted as $[\mathbf{L_M}, \mathbf{U_M}]$, which we prove to be true bounds and tighter than $[\mathbf{L_{IBP}}, \mathbf{U_{IBP}}]$ in expectation under a distribution of the network parameters $\mathbf{A}_1$ and $\mathbf{a}_2$. As such, we make two main theoretical findings. **(i)** We prove that $\mathbf{L_M}$ and $\mathbf{U_M}$ are true bounds in expectation, *i.e.* $\mathbb{E}_{\mathbf{A}_1, \mathbf{a}_2}[\mathbf{L_M}] \leq \mathbb{E}_{\mathbf{A}_1, \mathbf{a}_2}[\mathbf{L}_{\text{true}}]$ and $\mathbb{E}_{\mathbf{A}_1, \mathbf{a}_2}[\mathbf{U_M}] \geq \mathbb{E}_{\mathbf{A}_1, \mathbf{a}_2}[\mathbf{U}_{\text{true}}]$ hold for a sufficiently large input dimension $n$. **(ii)** We prove that $[\mathbf{L_M}, \mathbf{U_M}]$ can be arbitrarily tighter than the loose bounds $[\mathbf{L_{IBP}}, \mathbf{U_{IBP}}]$ in expectation, as the number of hidden nodes $k$ increases.

**Analysis.** To derive $\mathbf{L_M}$ and $\mathbf{U_M}$, we study the bounds of the following function first:

$$\tilde{g}(\tilde{\mathbf{x}}) = \mathbf{a}_2^\top \mathbf{M}(\mathbf{A}_1\tilde{\mathbf{x}} + \mathbf{b}_1) + \mathbf{b}_2 = \mathbf{a}_2^\top \mathbf{M}\mathbf{A}_1\tilde{\mathbf{x}} + \mathbf{a}_2^\top \mathbf{M}\mathbf{b}_1 + \mathbf{b}_2. \tag{2}$$

Note that $\tilde{g}$ is very similar to the Affine-ReLU-Affine map captured by $g$ with the ReLU replaced by a diagonal matrix $\mathbf{M}$ constructed as follows. If we denote $\mathbf{u}_1 = \mathbf{A}_1\mathbf{x} + \mathbf{b}_1 + \epsilon|\mathbf{A}_1|\mathbf{1}_n$ as the upper bound resulting from the propagation of the input bounds $[\mathbf{x} - \epsilon\mathbf{1}_n, \mathbf{x} + \epsilon\mathbf{1}_n]$ through the first affine map $(\mathbf{A}_1, \mathbf{b}_1)$, then we have $\mathbf{M} = \text{diag}(\mathbb{1}\{\mathbf{u}_1 \geq \mathbf{0}_k\})$ where $\mathbb{1}$ is an indicator function. In other words, $\mathbf{M}_{ii} = 1$ when the $i^{\text{th}}$ element of $\mathbf{u}_1$ is non-negative and zero otherwise. Note that for a given $\mathbf{u}_1$, $\tilde{g}(\tilde{\mathbf{x}})$ is an affine function with the following output interval bounds for $\tilde{\mathbf{x}} \in [\mathbf{x} - \epsilon\mathbf{1}_n, \mathbf{x} + \epsilon\mathbf{1}_n]$:

$$\mathbf{L_M}, \mathbf{U_M} = \mathbf{a}_2^\top \mathbf{M}\mathbf{A}_1\mathbf{x} + \mathbf{a}_2^\top \mathbf{M}\mathbf{b}_1 + \mathbf{b}_2 \mp \epsilon|\mathbf{a}_2^\top \mathbf{M}\mathbf{A}_1|\mathbf{1}_n \tag{3}$$

To compare $\mathbf{L_M}$ and $\mathbf{U_M}$ to $\mathbf{L}_{\text{true}}$ and $\mathbf{U}_{\text{true}}$, respectively, and since having access to $\mathbf{L}_{\text{true}}$ and $\mathbf{U}_{\text{true}}$ is not feasible, we make the following mild key assumption.

**Assumption 1.** *(Key Assumption). Consider an $\ell_\infty$ bounded uniform random variable $\tilde{\mathbf{x}}$, $\tilde{\mathbf{x}} \in [\mathbf{x} - \epsilon\mathbf{1}_n, \mathbf{x} + \epsilon\mathbf{1}_n]$ where $\mathbf{A}_1$ and $\mathbf{a}_2$ have elements that are i.i.d. Gaussian of zero mean and $\sigma_{\mathbf{A}_1}$ and $\sigma_{\mathbf{a}_2}$ standard deviations, then there exists a sufficiently large $m$, such that:*

$$\mathbb{E}_{\mathbf{A}_1, \mathbf{a}_2}[\mathbf{L}_{\text{true}}] \geq \mathbf{L}_{\text{approx}} = \mathbb{E}_{\mathbf{A}_1, \mathbf{a}_2, \tilde{\mathbf{x}}}[g(\tilde{\mathbf{x}})] - m\sqrt{Var_{\mathbf{A}_1, \mathbf{a}_2, \tilde{\mathbf{x}}}[g(\tilde{\mathbf{x}})]},$$

$$\mathbb{E}_{\mathbf{A}_1, \mathbf{a}_2}[\mathbf{U}_{\text{true}}] \leq \mathbf{U}_{\text{approx}} = \mathbb{E}_{\mathbf{A}_1, \mathbf{a}_2, \tilde{\mathbf{x}}}[g(\tilde{\mathbf{x}})] + m\sqrt{Var_{\mathbf{A}_1, \mathbf{a}_2, \tilde{\mathbf{x}}}[g(\tilde{\mathbf{x}})]}.$$

Under Assumption 1 and to show that $\mathbf{L_M}$ and $\mathbf{U_M}$ are true bounds in expectation, it is sufficient to show that $\mathbf{L}_{\text{approx}} \geq \mathbb{E}_{\mathbf{A}_1, \mathbf{a}_2}[\mathbf{L_M}]$ and similarly for $\mathbf{U_M}$. However, since it is generally difficult to compute the right hand sides of Assumption 1, they can be well approximated by Lyapunov Central Limit Theorem. More formally, follows the two propositions.

**Proposition 1.** *For $\mathbf{a} \in \mathbb{R}^n \sim \mathcal{N}(\mathbf{0}, \sigma_a^2\mathbf{I})$ and a uniform random vector $\tilde{\mathbf{x}} \sim \mathcal{U}[\mathbf{x} - \epsilon\mathbf{1}_n, \mathbf{x} + \epsilon\mathbf{1}_n]$ where both $\mathbf{a}$ and $\tilde{\mathbf{x}}$ are independent, we have that Lyapunov Central Limit Theorem holds such that*

$$\frac{1}{s_n} \sum_{i=1}^n (\tilde{x}_i a_i - \mathbb{E}[a_i \tilde{x}_i]) \to^d \mathcal{N}(0, 1), \quad \text{where} \ \ s_n^2 = Var\left( \sum_{i=1}^n (\tilde{x}_i a_i - \mathbb{E}[a_i \tilde{x}_i]) \right)$$

*where $\to^d$ indicates convergence in distribution.*

**Proposition 2.** *For a random matrix $\mathbf{A}_1$ with i.i.d. Gaussian elements of zero mean and $\sigma_{\mathbf{A}_1}$ standard deviation and a uniform random vector $\tilde{\mathbf{x}} \sim \mathcal{U}[\mathbf{x} - \epsilon\mathbf{1}_n, \mathbf{x} + \epsilon\mathbf{1}_n]$ we have that $Covariance(\mathbf{A}\tilde{\mathbf{x}}) = (\frac{1}{3}\epsilon^2\sigma_{\mathbf{A}_1}^2 n + \sigma_{\mathbf{A}_1}^2 trace(\mathbf{x}\mathbf{x}^\top))\mathbf{I}$.*

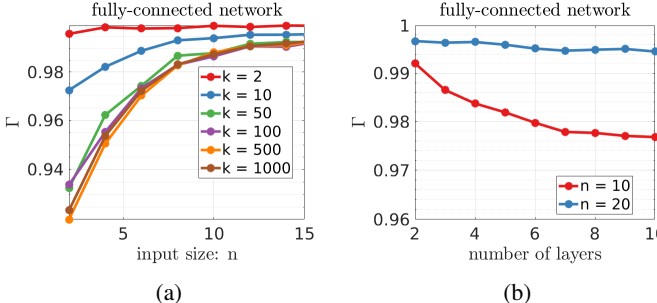

(a)           (b)

Figure 1: **True Bounds in Expectation.** As shown in Figure 1a, our proposed interval bounds $[\mathbf{L_M}, \mathbf{U_M}]$, as predicted by Theorem 1, get closer to being a true super set to the true interval bounds $[\mathbf{L}_{\text{true}}, \mathbf{U}_{\text{true}}]$ estimated by Monte-Carlo Sampling as the input dimension $n$ increases regardless of the number of hidden nodes. Figure 1b shows that a similar behaviour is present even under varying network depth.

Following Propositions 1 and 2, we have that for a sufficiently large input $n$:

$$\mathbf{L}_{\text{approx}}, \mathbf{U}_{\text{approx}} \approx \mathbb{E}_{\mathbf{a}_2,\tilde{\mathbf{y}}}\left[\mathbf{a}_2^\top \max\left(\tilde{\mathbf{y}},\mathbf{0}\right)+b_2\right] \mp m\sqrt{\text{Var}_{\mathbf{a}_2,\tilde{\mathbf{y}}}\left[\mathbf{a}_2^\top \max\left(\tilde{\mathbf{y}},\mathbf{0}\right)+b_2\right]}, \quad (4)$$

where the output of the first affine layer $\mathbf{A}_1\tilde{\mathbf{x}} + \mathbf{b}_1$ is approximated by Lyapunov Central Limit Theorem as $\tilde{\mathbf{y}} \sim \mathcal{N}(\mathbf{b}_1, (\frac{1}{3}\epsilon^2\sigma_{\mathbf{A}_1}^2 n + \sigma_{\mathbf{A}_1}^2\text{trace}(\mathbf{x}\mathbf{x}^\top))\mathbf{I})$.

**Theorem 1.** *(True Bounds in Expectation) Let Assumption 1 hold. For large input dimension $n$,*
$$\mathbb{E}_{\mathbf{A}_1,\mathbf{a}_2}\left[\mathbf{L_M}\right] \le \mathbb{E}_{\mathbf{A}_1,\mathbf{a}_2}\left[\mathbf{L}_{\text{true}}\right] \quad and \quad \mathbb{E}_{\mathbf{A}_1,\mathbf{a}_2}\left[\mathbf{U}_{\text{true}}\right] \le \mathbb{E}_{\mathbf{A}_1,\mathbf{a}_2}\left[\mathbf{U_M}\right]. \quad (5)$$

Theorem 1 states that the interval bounds for function $\tilde{g}$ are simply looser bounds to the function of interest $g$ in expectation under a plausible distribution of $\mathbf{A}_1$ and $\mathbf{a}_2$. Now, we investigate the tightness of these bounds as compared to the IBP bounds $[\mathbf{L}_{\text{IBP}}, \mathbf{U}_{\text{IBP}}]$.

**Theorem 2.** *(Tighter Bounds in Expectation) Consider an $\ell_\infty$ bounded uniform random variable input $\tilde{\mathbf{x}} \in [\mathbf{x} - \epsilon\mathbf{1}_n, \mathbf{x} + \epsilon\mathbf{1}_n]$ to a block of layers in the form of Affine-ReLU-Affine (parameterized by $\mathbf{A}_1, \mathbf{b}_1, \mathbf{a}_2$ and $\mathbf{b}_2$ for the first and second affine layers respectively) and $\mathbf{a}_2 \sim \mathcal{N}(\mathbf{0}, \sigma_{\mathbf{a}_2}\mathbf{I})$. Under the assumption that $\frac{1}{\sqrt{2\pi}}\mathbf{x}_j\mathbf{1}_k^\top\mathbf{A}_1(:,j) + \frac{1}{2n}\mathbf{1}_k^\top\mathbf{b}_1 \ge \epsilon\left(\|\mathbf{A}_1(:,j)\|_2 - \frac{1}{\sqrt{2\pi}}\|\mathbf{A}_1(:,j)\|_1\right)\ \forall j$, we have: $\mathbb{E}_{\mathbf{a}_2}\left[(\mathbf{U}_{\text{IBP}} - \mathbf{L}_{\text{IBP}}) - (\mathbf{U_M} - \mathbf{L_M})\right] \ge 0$.*

Theorem 2 states that under some assumptions on $\mathbf{A}_1$ and under a plausible distribution for $\mathbf{a}_2$, our proposed interval width can be much smaller than the IBP interval width, *i.e.* our proposed intervals are much tighter than the IBP intervals in expectation. Next, we show that the inequality assumption in Theorem 2 is very mild. In fact, a wide range of $(\mathbf{A}_1, \mathbf{b}_1)$ satisfy it, and the following proposition gives an example that does so in expectation.

**Proposition 3.** *For a random matrix $\mathbf{A}_1 \in \mathbb{R}^{k \times n}$ with i.i.d elements $\mathbf{A}_1(i,j) \sim \mathcal{N}(0,1)$, then*

$$\mathbb{E}_{\mathbf{A}_1}\left(\|\mathbf{A}_1(:,j)\|_2 - \frac{1}{\sqrt{2\pi}}\|\mathbf{A}_1(:,j)\|_1\right) = \sqrt{2}\frac{\Gamma\left(\frac{k+1}{2}\right)}{\Gamma\left(\frac{k}{2}\right)} - k\sqrt{\frac{2}{\pi}} \approx \sqrt{k}\left(1 - \sqrt{\frac{2}{\pi}}\sqrt{k}\right).$$

Proposition 3 implies that as the number of hidden nodes $k$ increases, the expectation of the right hand side of the inequality assumption in Theorem 2 grows more negative, while the left hand side of the inequality is zero in expectation when $\mathbf{b}_1 \sim \mathcal{N}(\mathbf{0}_k, \mathbf{I})$. In other words, for Gaussian zero-mean weights $(\mathbf{A}_1, \mathbf{b}_1)$ and with a large enough number of hidden nodes $k$, the assumption is satisfied. All proofs and detailed analyses are provided in the **appendix**.

**Comment on the Assumptions on $\mathbf{A}_1$ and $\mathbf{a}_2$.** Generally speaking, our proposed bounds $[\mathbf{L_M}, \mathbf{U_M}]$ computed using Equation 3 can be very loose and much worse compared to IBP if the network weights $\mathbf{A}_1, \mathbf{a}_2$ do not follow the Gaussian assumption. We leave for the **appendix** an example to such failure. However, while the Gaussian i.i.d. assumption can be strong for general networks trained on real data, it is not far from being reasonable due to commonly accepted training procedures. This is since it is common to regularize network weights while training deep neural networks with an $\ell_2$ regularizer encouraging the weights to follow a zero mean Gaussian distribution not to mention that networks in many cases are initialized in such manner. We dwell on this further in the **appendix** where we show the histogram of the weights of networks trained on real data (MNIST, CIFAR10, CIFAR100). with and without $\ell_2$ regularization See **appendix** for details.

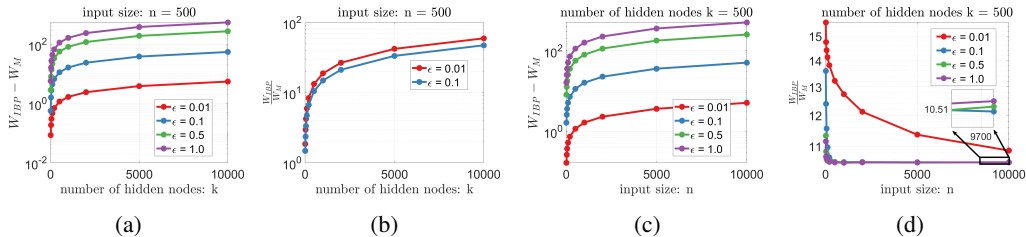

Figure 2: **Tighter than IBP with Varying Input Size and Hidden Nodes.** We show a bound tightness comparison between our proposed interval bounds and those of IBP by comparing the difference and ratio of their interval lengths with varying $k$, $n$, and $\epsilon$ for a two-layer network. The proposed bounds are significantly tighter than IBP, as predicted by Theorem 2.

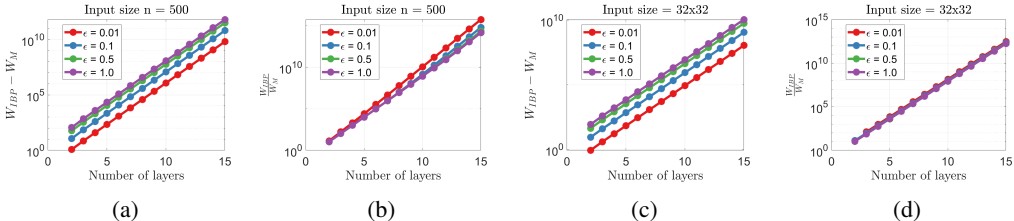

Figure 3: **Tighter than IBP in Deeper Networks.** We show a bound tightness comparison between our proposed interval bounds and those of IBP by varying the number of layers for several choices of $\epsilon$. The proposed bounds are significantly tighter than IBP.

### 3.3 Extending Our Expected Tight Bounds to Deeper Networks

To extend our proposed bounds to networks deeper than a two-layer block, we simply apply our bound procedure described in Section 3.2, recursively for every block. In particular, consider an $L$-layer neural network defined as $f(\mathbf{x}) = \mathbf{A}_L \text{ReLU}(\mathbf{A}_{L-1}\text{ReLU}(\cdots \mathbf{A}_2\text{ReLU}(\mathbf{A}_1\mathbf{x})))$ and an $\epsilon$-$\ell_\infty$ norm bounded input centered at $\mathbf{x}$, *i.e.* $\tilde{\mathbf{x}} \in [\mathbf{x} - \epsilon\mathbf{1}_n, \mathbf{x} + \epsilon\mathbf{1}_n]$. Without loss of generality, we assume $f$ is bias-free for ease of notation. Then, the output lower and upper bounds of $f$ are $\mathbf{L_M} = \mathbf{G}_{L-1}\mathbf{x} - \epsilon|\mathbf{G}_{L-1}|\mathbf{1}_n$ and $\mathbf{U_M} = \mathbf{G}_{L-1}\mathbf{x} + \epsilon|\mathbf{G}_{L-1}|\mathbf{1}_n$, respectively. Here, $\mathbf{G}_{L-1}$ is a linear map that can be obtained recursively as follows:

$$\mathbf{G}_i = \mathbf{A}_{i+1}\mathbf{M}_i\mathbf{G}_{i-1} \ \ (\text{with} \ \ \mathbf{G}_0 = \mathbf{A}_1) \ \ and \ \ \mathbf{M}_i = \text{diag}\left(\mathbb{1}\left\{(\mathbf{G}_{i-1}\mathbf{x} + \epsilon|\mathbf{G}_{i-1}|\mathbf{1}_n) \geq \mathbf{0}\right\}\right) \ \ (6)$$

Note that $\mathbf{G}_{i-1}\mathbf{x} + \epsilon|\mathbf{G}_{i-1}|\mathbf{1}_n$ is the output upper bound through a linear layer parameterized by $\mathbf{G}_{i-1}$ for input $\tilde{\mathbf{x}}$ as in (1). With this blockwise propagation, the output interval bounds of $f$ are now estimated by the output intervals of $\tilde{f}(\tilde{\mathbf{x}}) = \mathbf{G}_{L-1}\tilde{\mathbf{x}}$.

## 4 Experiments

**True Bounds in Expectation.** Here, we validate Theorem 1 with several controlled experiments. For a network $g(\tilde{\mathbf{x}}) = \mathbf{A}_2 \max(\mathbf{A}_1\tilde{\mathbf{x}} + \mathbf{b}_1, \mathbf{0}) + \mathbf{b}_2$ that has true bounds $[\mathbf{L}_{\text{true}}, \mathbf{U}_{\text{true}}]$ for $\tilde{\mathbf{x}} \in [\mathbf{x} - \epsilon\mathbf{1}_n, \mathbf{x} + \epsilon\mathbf{1}_n]$, we empirically show that our proposed bounds $[\mathbf{L_M}, \mathbf{U_M}]$, under the mild assumptions of Theorem 1, indeed are true, *i.e.* they are a super set to $[\mathbf{L}_{\text{true}}, \mathbf{U}_{\text{true}}]$ in expectation. We also verify this as a function of the network input dimension $n$ (as predicted by Theorem 1).

We start by constructing a network $g$ where the biases $\mathbf{b}_1 \in \mathbb{R}^k$ and $b_2 \in \mathbb{R}$ are initialized following the default Pytorch initialization (Paszke et al., 2017). As for the elements of the weight matrices $\mathbf{A}_1 \in \mathbb{R}^{k \times n}$ and $\mathbf{A}_2 \in \mathbb{R}^{1 \times k}$, they are sampled from $\mathcal{N}(0, 1/\sqrt{n})$ and $\mathcal{N}(0, 1/\sqrt{k})$, respectively. We estimate $\mathbf{L}_{\text{true}}$ and $\mathbf{U}_{\text{true}}$ by taking the minimum and maximum of $10^6 + 2^n$ Monte-Carlo evaluations of $g$. For a given $\mathbf{x} \sim \mathcal{N}(\mathbf{0}_n, \mathbf{I})$ and with $\epsilon = 0.1$, we uniformly sample $10^6$ examples from the interval $[\mathbf{x} - \epsilon\mathbf{1}_n, \mathbf{x} + \epsilon\mathbf{1}_n]$. We also sample all $2^n$ corners of the hyper cube $[\mathbf{x} - \epsilon\mathbf{1}_n, \mathbf{x} + \epsilon\mathbf{1}_n]^n$. To show that the proposed interval $[\mathbf{L_M}, \mathbf{U_M}]$ is a super set of $[\mathbf{L}_{\text{true}}, \mathbf{U}_{\text{true}}]$ (*i.e.* they are true bounds), we evaluate the length of the intersection of the two intervals over the length of the true interval defined as $\Gamma = |[\mathbf{L_M}, \mathbf{U_M}] \cap [\mathbf{L}_{\text{true}}, \mathbf{U}_{\text{true}}]|/|[\mathbf{L}_{\text{true}}, \mathbf{U}_{\text{true}}]|$. Note that $\Gamma = 1$ if and only if $[\mathbf{L_M}, \mathbf{U_M}]$ is a super set of $[\mathbf{L}_{\text{true}}, \mathbf{U}_{\text{true}}]$. For a given $n$, we conduct this experiment $10^3$ times with varying $\mathbf{A}_1$, $\mathbf{A}_2$, $\mathbf{b}_1$, $\mathbf{b}_2$ and $\mathbf{x}$ and report the average $\Gamma$. Then, we run this for a varying number of input size $n$ and a varying number of hidden nodes $k$, as reported in Figure 1a. As predicted by Theorem 1,

Table 1: **True and Tight Bounds on Real Networks.** Table shows that our bounds are a super set to true bounds computed with an exact MIP solver. Moreover, they are much tighter than bounds estimated by IBP.

| | $\epsilon$ | $\Gamma$ | $\Gamma_{\min}$ | $\Gamma_{\max}$ | | $\epsilon$ | $W_{\text{IBP}} - W_{\text{M}}$ | $W_{\text{IBP}}/W_{\text{M}}$ |
|---|---|---|---|---|---|---|---|---|
| Small MNIST | 0.01 | $1.0 \pm 0$ | 1.0 | 1.0 | Small MNIST | 0.01 | 644.322 | 17.391 |
| | 0.02 | $1.0 \pm 0$ | 1.0 | 1.0 | | 0.02 | 1381.980 | 15.270 |
| | 0.03 | $0.97 \pm 0.088$ | 0.635 | 1.0 | | 0.03 | 2255.397 | 14.555 |

Figure 4: **Qualitative Results.** We plot visualizations of the output polytope of a 20-100-100-100-100-2 network through Monte-Carlo evaluations of the network with a uniform random input with varying $\epsilon$. We also plot our proposed bounds $[\mathbf{L}_{\text{M}}, \mathbf{U}_{\text{M}}]$ in red. Each row is for a given $\epsilon$ with different randomized weights for the network. As for the IBP bounds $[\mathbf{L}_{\text{IBP}}, \mathbf{U}_{\text{IBP}}]$, they were omitted as they were significantly larger. For example, for the first figure with $\epsilon = 0.05$, IBP bounds are $[-43.7, 32.9]$ for the x-axis and $[-47.8, 37.0]$ for the y-axis.

Figure 1a demonstrates that as $n$ increases, the proposed interval will be more likely to be a super set of the true interval, regardless of the number of hidden nodes $k$. Note that networks that are as wide as $k = 1000$, require no more than $n = 15$ input dimensions for the proposed intervals to be a super set of the true intervals. In practice, $n$ is much larger than that, *e.g.* $n \approx 3 \times 10^3$ in CIFAR10.

In Figure 1b, we empirically show that the above behavior also holds for deeper networks. We propagate the bounds blockwise as discussed in Section 3.3 and conduct similar experiments on fully-connected networks. We construct networks with varying depth, where each layer has the same number of nodes equal to the input dimension $k = n$. These results indeed suggest that the proposed bounds are true bounds and are more likely so with larger input dimensions. Here, $n = 20$ performs better than $n = 10$ across different network depths.

**Tighter Bounds in Expectation.** Here, we experimentally affirm that our bounds can be much tighter than IBP bounds (Gowal et al., 2019). In particular, we validate Theorem 2 by comparing the interval length of our proposed bounds, $W_{\text{M}} = \mathbf{U}_{\text{M}} - \mathbf{L}_{\text{M}}$, with that of IBP, $W_{\text{IBP}} = \mathbf{U}_{\text{IBP}} - \mathbf{L}_{\text{IBP}}$, on networks with functional form $g$. We compute both the difference and ratio of widths for varying values of $k$, $n$, and $\epsilon$. Figure 2 reports the average width difference and ratio over $10^3$ runs in a similar setup to the previous section. Figures 2a and 2b show that the proposed bounds indeed get tighter than IBP, as $k$ increases across all $\epsilon$ values (as predicted by Theorem 2). Note that we show tightness results for $\epsilon = \{0.01, 0.1\}$ in Figure 2b as the performance of $\epsilon = \{0.5, 1.0\}$ was very similar to $\epsilon = 0.1$. Similar improvement occurs with increasing $n$, as in Figures 2c and 2d.

We also compare the tightness of our bounds to those of IBP with increasing depth for both fully-connected networks (refer to Figures 3a and 3b) and convolutional networks (refer to Figures 3c and 3d). For all fully-connected networks, we take $n = k = 500$. Our proposed bounds get consistently tighter as the network depth increases over all choices of $\epsilon$. In particular, the proposed bounds can be more than $10^6$ times tighter than IBP for a 10 layer DNN. A similar observation can also be made for convolutional networks, where it is expensive to compute our bounds using the procedure in Section 3.3, so instead, we obtain matrices $\mathbf{M}_i$ using the easy-to-compute IBP upper bounds. Despite this relaxation, we still obtain very tight expected bounds. Note that this slightly modified approach reduces exactly to our bounds for two-layer networks.

**True and Tight Bounds on Real Networks.** Moreover, we also train small network, the architecture is similar to Gowal et al. (2019), on the MNIST dataset $\sim 99\%$. To show that our bounds are also true on real networks, and since the input dimension is too large for Monte-Carlo sampling, we use the MIP formulation by Tjeng & Tedrake (2019). We then report $\Gamma$ over varying testing $\epsilon$. Table 1 demonstrates that indeed even on real networks beyond two layers and without the Gaussian weight

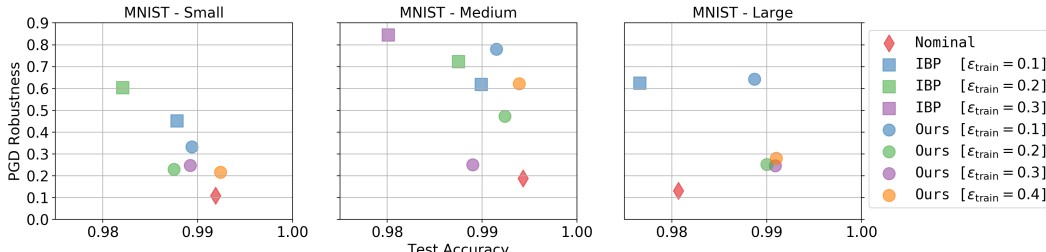

Figure 5: **Better Test Accuracy and Robustness on MNIST.** We compare the PGD robustness and test accuracy of three models (small, medium, and large) robustly trained on the MNIST dataset using our bounds and those robustly trained with IBP. We have trained both methods using four different $\epsilon_{\text{train}}$, but we eliminated all models with test accuracy lower than $97.5\%$. Our results demonstrate an impressive trade-off between accuracy and robustness and, in some cases (medium and large models), we excel in both.

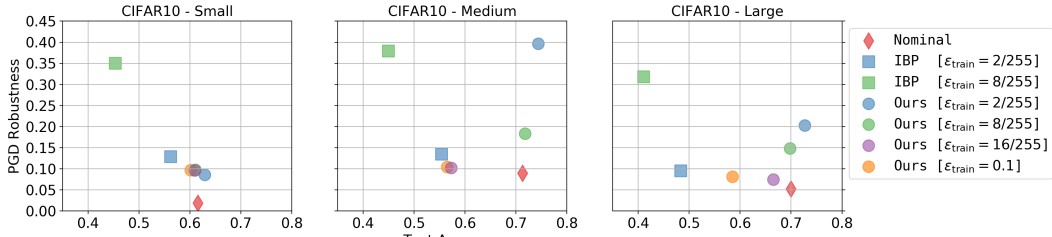

Figure 6: **Better Test Accuracy and Robustness on CIFAR10.** We compare the PGD robustness and test accuracy of three models (small, medium, and large) robustly trained on the CIFAR10 dataset using our bounds and those robustly trained with IBP. We eliminated all models with test accuracy lower than $40.0\%$. PGD robustness is averaged over multiple $\epsilon_{\text{test}}$ (refer to **appendix**).

distributed assumption, our bounds are still a super set to the true bounds computed with an MIP formulation. Moreover, Table 1 also demonstrates that our bounds are much tighter than IBP bounds. Note that the reported results are averaged over 100 random MNIST images.

**Qualitative Results.** Following previous work (Wong & Kolter, 2018; Gowal et al., 2019), we visualize examples of the proposed bounds in Figure 4 and compare them to the true ones for several choices of $\epsilon \in \{0.05, 0.1, 0.25\}$ and a random five-layer fully-connected network with architecture $n$-100-100-100-100-2. We also show the results of the Monte-Carlo sampling for an input size $n = 20$. More qualitative visualizations for different values of $n$ are in the **appendix**.

**Training Robust Networks.** Here, we conduct experiments showing that our expected bounds can be used to robustly train DNNs. We compare our method against models trained nominally (*i.e.* only the nominal training loss is used), and those trained robustly with IBP (Gowal et al., 2019). Given the well-known robustness-accuracy trade off, robust models are often less accurate. Therefore, we compare all methods using an accuracy vs. robustness scatter plot. Following prior work, we use Projected Gradient Descent (PGD) (Madry et al., 2018) to measure robustness. We use a loss function similar to the one proposed in (Gowal et al., 2019). In particular, we use $L = \ell(f_\theta(\mathbf{x}), \mathbf{y}_{\text{true}}) + \kappa\ell(\mathbf{z}, \mathbf{y}_{\text{true}})$, where $\ell, f_\theta(\mathbf{x}), \mathbf{y}_{\text{true}}$, and $\kappa$ are the cross-entropy loss, output logits, true class label, and regularization hyperparameter respectively. $\mathbf{z}$ represents the "adversarial" logits that combine the lower bound of the true label and the upper bound of all other labels, as in (Gowal et al., 2019). Nominal training occurs when $\kappa = 0$. Due to the tightness of our bounds, in contrast to IBP, we follow a standard training procedure that avoids the need to vary $\kappa$ or $\epsilon_{train}$ during training.

Specifically, we train three network models (small, medium, large) provided by Gowal et al. (2019) on both MNIST and CIFAR10. See **appendix** for more details. Following the same setup in (Gowal et al., 2019), we train all models with $\epsilon_{\text{train}} \in \{0.1, 0.2, 0.3, 0.4\}$ and $\epsilon_{\text{train}} \in \{2/255, 8/255, 16/255, 0.1\}$ on MNIST and CIFAR10, respectively. In all experiments, and for stronger baselines and fair comparison between IBP training and our bounds employed in training, we grid search over $\{0.1, 0.001, 0.0001\}$ learning rates and employ a temperature over the logits with a grid of $\{1, 1/5\}$ as in (Hinton et al., 2015) and report the best performing models for both. Then, we compute PGD robustness for every $\epsilon_{\text{train}}$ of every model for all $\epsilon_{\text{test}} \in \{0.1, 0.2, 0.3, 0.4\}$ for MNIST and for all $\epsilon_{\text{test}} \in \{2/255, 8/255, 16/255, 0.1\}$ for CIFAR10. To compare training methods, we compute the average PGD robustness over all $\epsilon_{\text{test}}$ and the test accuracy, and report them in a 2D scatter plot. We

report the performance results on MNIST and CIFAR10 for the small, medium, and large architectures in Figure 5. For all trained architectures, we only report the results for those that achieve at least a test accuracy of $97.5\%$ and $40\%$ on MNIST and CIFAR10, respectively; otherwise, it is an indication of failure in training. Interestingly, our training scheme can be used to train all architectures for all $\epsilon_{\text{train}}$. This is unlike IBP, which for example was only able to successfully train the large architecture with $\epsilon_{\text{train}} = 0.1$ on MNIST. Moreover, models trained with our bounds always achieve better PGD robustness than the nominally trained networks on all architectures while preserving similar if not higher accuracy (on large networks). Models trained with IBP achieve high robustness but their test accuracy is drastically affected. Several other experiments are left for the **appendix**.

## 5 CONCLUSION

In this work, we proposed new interval bounds that are tight, relatively cheap to compute, and true in expectation. We analytically showed that for a Affine-ReLU-Affine block with large input and hidden layer sizes, our bounds are true in expectation and can be several orders of magnitude tighter than the bounds obtained with IBP. We conduct extensive experiments verifying our theory, even for deep networks. As a result, we are able to train large models, with simple standard training routines while achieving excellent trade-off between accuracy and robustness.

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

**Proposition 1.** *For $\mathbf{a} \in \mathbb{R}^n \sim \mathcal{N}(\mathbf{0}, \sigma_a^2 \mathbf{I})$ and a uniform random vector $\tilde{\mathbf{x}} \sim \mathcal{U}[\mathbf{x} - \epsilon \mathbf{1}_n, \mathbf{x} + \epsilon \mathbf{1}_n]$ where both $\mathbf{a}$ and $\tilde{\mathbf{x}}$ are independent, we have that Lyapunov Central Limit Theorem holds such that*

$$\frac{1}{s_n} \sum_{i=1}^n (\tilde{x}_i a_i - \mathbb{E}[a_i \tilde{x}_i]) \to^d \mathcal{N}(0, 1), \text{ where } s_n^2 = Var\left(\sum_{i=1}^n (\tilde{x}_i a_i - \mathbb{E}[a_i \tilde{x}_i])\right)$$

*where $\to^d$ indicates convergence in distribution.*

*Proof.* The Lyapunov condition

$$\exists \delta > 0, \frac{1}{s_n^{2+\delta}} \sum_{i=1}^n \mathbb{E}\left[\left|\tilde{x}_i a_i - \mathbb{E}[a_i \tilde{x}_i]\right|^{2+\delta}\right] \to 0, \text{ as } n \to \infty \tag{7}$$

is sufficient for Lyanunov Central Limit Theorem to hold. Note that

$$s_n^2 = \sum_{i=1}^2 Var\left(\tilde{x}_i a_i\right) = \sum_{i=1}^n \mathbb{E}\left[a_i^2 \tilde{x}_i^2\right] = \sigma_a^2 \sum_{i=1}^n \left(\frac{\epsilon^2}{3} + x_i^2\right) = \sigma_a^2 \left(\frac{n\epsilon^2}{3} + \sum_{i=1}^n x_i^2\right). \tag{8}$$

Since for $\delta = 2$, we have that

$$\mathbb{E}[|a_i \tilde{x}_i|^{2+\delta}] = \int_{-\infty}^{\infty} \int_{x_i - \epsilon}^{x_i + \epsilon} a_i^2 \tilde{x}_i^2 \frac{1}{2\epsilon} \frac{1}{\sqrt{2\pi}\sigma_a} \exp\left(-\frac{a_i^2}{2\sigma_a^2}\right) da_i d\tilde{x}_i$$

$$= \frac{3\sigma_a^4}{2\epsilon} \int_{x_i - \epsilon}^{x_i + \epsilon} \tilde{x}_i^4 d\tilde{x}_i = \frac{3\sigma_a^4}{10\epsilon}\left[(x_i + \epsilon)^5 - (x_i - \epsilon)^5\right].$$

Thereafter, Lyanunov Central Limit Theorem with $\delta = 2$ is satisfied since

$$\lim_{n \to \infty} \frac{1}{s_n^4} \sum_{i=1}^n \mathbb{E}\left[\left|\tilde{x}_i a_i - \mathbb{E}[a_i \tilde{x}_i]\right|^{2+\delta}\right] = \lim_{n \to \infty} \frac{1}{s_n^4} \sum_{i=1}^n \mathbb{E}\left[\left|\tilde{x}_i a_i\right|^4\right]$$

$$= \lim_{n \to \infty} \frac{3 \sum_{i=1}^n (x_i + \epsilon)^5 - (x_i - \epsilon)^5}{10\epsilon \left(\frac{n\epsilon^2}{3} + \sum_{i=1}^n x_i^2\right)^2}$$

$$\leq \lim_{n \to \infty} \frac{3n\left((x_{\max} + \epsilon)^5 - (x_{\min} - \epsilon)^5\right)}{10\epsilon \left(\frac{n\epsilon^2}{3} + \sum_{i=1}^n x_i^2\right)^2} = 0$$

$\square$

**Proposition 2.** *For a random matrix $\mathbf{A}_1$ with i.i.d. Gaussian elements of zero mean and $\sigma_{\mathbf{A}_1}$ standard deviation and a uniform random vector $\tilde{\mathbf{x}} \sim \mathcal{U}[\mathbf{x} - \epsilon \mathbf{1}_n, \mathbf{x} + \epsilon \mathbf{1}_n]$ we have that Covariance $(\mathbf{A}\tilde{\mathbf{x}}) = \left(\frac{\epsilon^2 \sigma_{\mathbf{A}_1}^2 n}{3} + \sigma_{\mathbf{A}_1}^2 trace\left(\mathbf{x}\mathbf{x}^\top\right)\right) \mathbf{I}$.*

*Proof.* The former follows from the fact that

Covariance $(\mathbf{A}_1 \tilde{\mathbf{x}} + \mathbf{b}_1) =$ Covariance $(\mathbf{A}_1 \tilde{\mathbf{x}}) = \mathbb{E}\left[\mathbf{A}_1 \tilde{\mathbf{x}} \tilde{\mathbf{x}}^\top \mathbf{A}_1^\top\right] - \mathbb{E}\left[\mathbf{A}_1 \tilde{\mathbf{x}}\right]\left(\mathbb{E}\left[\mathbf{A}_1 \tilde{\mathbf{x}}\right]\right)^\top = \mathbb{E}\left[\mathbf{A}_1 \tilde{\mathbf{x}} \tilde{\mathbf{x}}^\top \mathbf{A}_1^\top\right]$

$$= \mathbb{E}_{\mathbf{A}_1}\left[\mathbf{A}_1 \mathbb{E}\left[\tilde{\mathbf{x}} \tilde{\mathbf{x}}^\top\right] \mathbf{A}_1^\top\right] = \mathbb{E}_{\mathbf{A}_1}\left[\mathbf{A}_1 \left(\text{Diag}\left(\frac{\epsilon^2}{3}\right) + \mathbf{x}\mathbf{x}^\top\right) \mathbf{A}_1^\top\right]$$

$$= \frac{\epsilon^2}{3} \mathbb{E}_{\mathbf{A}_1}\left[\mathbf{A}_1 \mathbf{A}_1^\top\right] + \mathbb{E}\left[\mathbf{A}_1 \mathbf{x}\mathbf{x}^\top \mathbf{A}_1^\top\right]$$

$$= \frac{\epsilon^2}{3} \mathbb{E}_{\mathbf{A}_1}\left[\mathbf{A}_1 \mathbf{A}_1^\top\right] + \sigma_{\mathbf{A}_1}^2 \text{trace}\left(\mathbf{x}\mathbf{x}^\top\right) \mathbf{I} = \left(\frac{\epsilon^2 \sigma_{\mathbf{A}_1}^2 n}{3} + \sigma_{\mathbf{A}_1}^2 \text{trace}\left(\mathbf{x}\mathbf{x}^\top\right)\right) \mathbf{I}$$

The last equality follows since:

$$\left(\mathbb{E}\left[\mathbf{A}_1 \mathbf{x}\mathbf{x}^\top \mathbf{A}_1^\top\right]\right)_{i,j} = \mathbb{E}\left[\mathbf{a}_i^\top \mathbf{x}\mathbf{x}^\top \mathbf{a}_j\right] = \text{trace}\left(\mathbf{x}\mathbf{x}^\top \mathbb{E}\left[\mathbf{a}_j \mathbf{a}_i^\top\right]\right) = \begin{cases} 0 & \text{if } i \neq j \\ \sigma_{\mathbf{A}_1}^2 \text{trace}\left(\mathbf{x}\mathbf{x}^\top\right) & \text{if } i = j \end{cases}$$

$\square$

**Theorem 1.** *(True Bounds in Expectation) Let Assumption 1 hold. We have that for large input dimension $n$,*

$$\mathbb{E}_{\mathbf{A}_1,\mathbf{a}_2}\left[\mathbf{L_M}\right] \leq \mathbb{E}_{\mathbf{A}_1,\mathbf{a}_2}\left[\mathbf{L}_{true}\right], \quad \mathbb{E}_{\mathbf{A}_1,\mathbf{a}_2}\left[\mathbf{U}_{true}\right] \leq \mathbb{E}_{\mathbf{A}_1,\mathbf{a}_2}\left[\mathbf{U_M}\right].$$

*Proof.*

$$\mathbf{L}_{\text{approx}} \approx \mathbb{E}_{\mathbf{a}_2,\tilde{\mathbf{y}}}\left[\mathbf{a}_2^\top \max\left(\tilde{\mathbf{y}},\mathbf{0}\right) + b_2\right] - m\sqrt{\mathrm{Var}_{\mathbf{a}_2,\tilde{\mathbf{y}}}\left[\mathbf{a}_2^\top \max\left(\tilde{\mathbf{y}},\mathbf{0}\right) + b_2\right]}$$

$$\overset{\text{①}}{=} b_2 - m\sqrt{\mathbb{E}_{\mathbf{a}_2}\left[\mathrm{Var}_{\tilde{\mathbf{y}}}\left(\mathbf{a}_2^\top \max\left(\tilde{\mathbf{y}},\mathbf{0}\right) + b_2|\mathbf{a}_2\right)\right] + \mathrm{Var}_{\mathbf{a}_2}\left(\mathbb{E}_{\tilde{\mathbf{y}}}\left[\mathbf{a}_2^\top \max\left(\tilde{\mathbf{y}},\mathbf{0}\right) + b_2|\mathbf{a}_2\right]\right)}$$

$$= b_2 - m\left(\mathbb{E}_{\mathbf{a}_2}\left[\left(\mathbf{a}_2^\top \odot \mathbf{a}_2^\top\right)\left(\mathbb{E}_{\tilde{\mathbf{y}}}\left[\max^2\left(\tilde{\mathbf{y}},\mathbf{0}\right)\right] - \left(\mathbb{E}_{\tilde{\mathbf{y}}}\left[\max\left(\tilde{\mathbf{y}},\mathbf{0}\right)\right]\right)^2\right)\right] + \mathrm{Var}_{\mathbf{a}_2}\left(\mathbb{E}_{\tilde{\mathbf{y}}}\left[\mathbf{a}_2^\top \max\left(\tilde{\mathbf{y}},\mathbf{0}\right) + b_2|\mathbf{a}_2\right]\right)\right)^{\frac{1}{2}}$$

$$= b_2 - m\left(\left[\sum_{i=1}^{k}\sigma^2_{\mathbf{a}_2}\left(\mathbb{E}_{\tilde{\mathbf{y}}}\left[\max^2\left(\tilde{\mathbf{y}},\mathbf{0}\right)\right]\right)_i - \sum_{i=1}^{k}\sigma^2_{\mathbf{a}_2}\left(\mathbb{E}_{\tilde{\mathbf{y}}}\left[\max\left(\tilde{\mathbf{y}},\mathbf{0}\right)\right]\right)^2_i\right] + \sum_{i=1}^{k}\sigma^2_{\mathbf{a}_2}\left(\mathbb{E}_{\tilde{\mathbf{y}}}\left[\max\left(\tilde{\mathbf{y}},\mathbf{0}\right)\right]\right)^2_i\right)^{\frac{1}{2}}$$

$$= b_2 - m\sigma_{\mathbf{a}_2}\sqrt{\sum_{i=1}^{k}\left(\mathbb{E}_{\tilde{\mathbf{y}}}\left[\max^2\left(\tilde{\mathbf{y}},\mathbf{0}\right)\right]\right)_i}$$

$$\overset{\text{②}}{=} b_2 - m\sigma_{\mathbf{a}_2}\underbrace{\sqrt{\left(\mathbf{b}_1^{2i} + \sigma^2_{\tilde{\mathbf{y}}}\right)\odot\Phi\left(\mathbf{b}_1^i \oslash \sigma_{\tilde{\mathbf{y}}}\right) + \left(\mathbf{b}_1^i \odot \sigma_{\tilde{\mathbf{y}}} \odot \phi\left(\mathbf{b}_1^i \oslash \sigma_{\tilde{\mathbf{y}}}\right)\right)}}_{\Psi}.$$

Note that ① follows by total expectation and total variance on the two terms, respectively. Lastly, ② follows from the closed form expression derived in Bibi et al. (2018) where $\Phi$ and $\phi$ are the normal cumulative and probability Gaussian density functions, respectively. Note that $\tilde{\mathbf{y}} \sim \mathcal{N}\left(\mathbf{b}_1, \left(\frac{\epsilon^2\sigma^2_{\mathbf{A}_1}n}{3} + \sigma^2_{\mathbf{A}_1}\mathrm{trace}\left(\mathbf{x}\mathbf{x}^\top\right)\right)\mathbf{I}\right)$ and that $\sigma^2_{\tilde{\mathbf{y}}} = \left(\frac{\epsilon^2\sigma^2_{\mathbf{A}_1}n}{3} + \sigma^2_{\mathbf{A}_1}\mathrm{trace}\left(\mathbf{x}\mathbf{x}^\top\right)\right)$.

$$\mathbf{L}_{\text{approx}} - \mathbb{E}_{\mathbf{A}_1,\mathbf{a}_2}\left[\mathbf{L_M}\right]$$

$$\approx b_2 - m\sigma_{\mathbf{a}_2}\Psi - \mathbb{E}_{\mathbf{A}_1,\mathbf{a}_2}\left[\mathbf{a}_2^\top\mathbf{M}\left(\mathbf{A}_1\mathbf{x} + \mathbf{b}_1\right) + b_2 - \epsilon|\mathbf{a}_2^\top\mathbf{MA}_1|\mathbf{1}\right]$$

$$= \mathbb{E}_{\mathbf{A}_1,\mathbf{a}_2}\left[\epsilon|\mathbf{a}_2^\top\mathbf{MA}_1|\mathbf{1}\right] - m\sigma_{\mathbf{a}_2}\Psi$$

$$= \epsilon\mathbb{E}_{\mathbf{A}_1}\left[\sum_{j=1}^{n}\mathbb{E}_{\mathbf{a}_2}\left[|\mathbf{a}_2^\top\mathbf{MA}_1(:,j)||\mathbf{A}_1\right]\right] - m\sigma_{\mathbf{a}_2}\Psi$$

$$\overset{\text{①}}{=} \epsilon\sqrt{\frac{2}{\pi}}\mathbb{E}_{\mathbf{A}_1}\left[\sum_{j=1}^{n}\sqrt{\mathrm{Var}_{\mathbf{a}_2}\left(\mathbf{a}_2^\top\mathbf{MA}_1(:,j)\right)}\right] - m\sigma_{\mathbf{a}_2}\Psi$$

$$= \epsilon\sigma_{\mathbf{a}_2}\sqrt{\frac{2}{\pi}}\mathbb{E}_{\mathbf{A}_1}\left[\sum_{j=1}^{n}\sqrt{\mathbf{A}_1(:,j)^\top\mathbf{MA}_1(:,j)}\right] - m\sigma_{\mathbf{a}_2}\Psi$$

$$= \epsilon\sigma_{\mathbf{a}_2}\sqrt{\frac{2}{\pi}}\mathbb{E}_{\mathbf{A}_1}\left[\sum_{j=1}^{n}\sqrt{\sum_{i=1}^{k}\mathbf{A}_1(i,j)^2\mathbb{1}\left\{\mathbf{u}_1^i \geq 0\right\}}\right] - m\sigma_{\mathbf{a}_2}\Psi$$

$$= \epsilon\sigma_{\mathbf{a}_2}\sqrt{\frac{2}{\pi}}\mathbb{E}_{|S|}\left[\mathbb{E}_{\mathbf{A}_1}\left[\sum_{j=1}^{n}\sqrt{\sum_{i\in S}\mathbf{A}_1(i,j)^2}\,\Big|\,|S|\right]\right] - m\sigma_{\mathbf{a}_2}\Psi$$

Note that ① follows from the mean of a folded Gaussian. The last equality follows by taking the total expectation where $S$ is the set of indices where $\mathbf{u}_1^i \geq \mathbf{0}$ for all $i \in S$. Since $\mathbf{u}_1$ is random, then

$|S|$ is also random. Therefore, one can reparametrize the sum and thus we have

$$\epsilon\sigma_{\mathbf{a}_2}\sqrt{\frac{2}{\pi}}\mathbb{E}_{|S|}\left[\mathbb{E}_{\mathbf{A}_1}\left[\sum_{j=1}^{n}\sqrt{\sum_{i\in S}\mathbf{A}_1(i,j)^2}\right]\Big|\,|S|\right] = \frac{2\epsilon\sigma_{\mathbf{A}_1}\sigma_{\mathbf{a}_2}n}{\sqrt{\pi}}\mathbb{E}_{|S|}\left[\frac{\Gamma\left(\frac{|S|+1}{2}\right)}{\Gamma\left(\frac{|S|}{2}\right)}\right]$$

$$\approx \frac{2\epsilon\sigma_{\mathbf{A}_1}\sigma_{\mathbf{a}_2}n}{\sqrt{\pi}}\mathbb{E}_{|S|}\left[\sqrt{\frac{|S|}{2}}\right]$$

$$\geq \frac{2\epsilon\sigma_{\mathbf{A}_1}\sigma_{\mathbf{a}_2}n}{\sqrt{\pi}}\sqrt{\frac{k}{2}}$$

The approximation follows from stirlings formula $\Gamma(x+1/2)/\Gamma(x/2) \approx \sqrt{x/2}$ for large x where the last inequality follows since $\mathbb{E}\left[\sqrt{|S|}\right] \leq k$.

$$\mathbf{L}_{\text{approx}} - \mathbb{E}_{\mathbf{A}_1,\mathbf{a}_2}\left[\mathbf{L_M}\right] \geq \underbrace{\frac{2\epsilon\sigma_{\mathbf{A}_1}\sigma_{\mathbf{a}_2}n}{\sqrt{\pi}}\sqrt{\frac{k}{2}}}_{①}$$

$$\underbrace{- m\sigma_{\mathbf{a}_2}\sqrt{\sum_{i=1}^{k}\left(\mathbf{b}_1^{2i}+\sigma_{\tilde{\mathbf{y}}}^2\right)\odot\Phi\left(\mathbf{b}_1^i\oslash\sigma_{\tilde{\mathbf{y}}}\right)+\left(\mathbf{b}_1^i\odot\sigma_{\tilde{\mathbf{y}}}\odot\phi\left(\mathbf{b}_1^i\oslash\sigma_{\tilde{\mathbf{y}}}\right)\right)}}_{②}$$

Note that ① is $\mathcal{O}(n)$ while ② is $\mathcal{O}(\sqrt{n})$. Thus, for sufficiently large input dimension $n$ we have that $\mathbf{L}_{\text{approx}} \geq \mathbb{E}_{\mathbf{A}_1,\mathbf{a}_2}[\mathbf{L_M}]$ and since by construction $\mathbb{E}_{\mathbf{A}_1,\mathbf{a}_2}\left[\mathbf{L}_{\text{true}}\right] \geq \mathbf{L}_{\text{approx}}$ which completes the proof. Note that a symmetric argument can be applied to show that $\mathbb{E}_{\mathbf{A}_1,\mathbf{a}_2}\left[\mathbf{U_M}\right] \geq \mathbb{E}_{\mathbf{A}_1,\mathbf{a}_2}\left[\mathbf{U}_{\text{true}}\right]$. □

**Theorem 2.** *(Tighter Bounds in Expectation) Consider an $\ell_\infty$ bounded uniform random variable input $\tilde{\mathbf{x}}$, i.e. $\tilde{\mathbf{x}} \in [\mathbf{x} - \epsilon \mathbf{1}_n, \mathbf{x} + \epsilon \mathbf{1}_n]$, to a block of layers in the form of Affine-ReLU-Affine (parameterized by $\mathbf{A}_1, \mathbf{b}_1, \mathbf{a}_2$ and $\mathbf{b}_2$ for the first and second affine layers respectively) and $\mathbf{a}_2 \sim \mathcal{N}(\mathbf{0}, \sigma_{\mathbf{a}_2} \mathbf{I})$. Under the assumption that $\frac{1}{\sqrt{2\pi}} \mathbf{x}_j \mathbf{1}_k^\top \mathbf{A}_1(:,j) + \frac{1}{2n} \mathbf{1}_k^\top \mathbf{b}_1 \geq \epsilon \left( \|\mathbf{A}_1(:,j)\|_2 - \frac{1}{\sqrt{2\pi}} \|\mathbf{A}_1(:,j)\|_1 \right) \forall j$, we have: $\mathbb{E}_{\mathbf{a}_2} \left[ (\mathbf{U}_{IBP} - \mathbf{L}_{IBP}) - (\mathbf{U}_\mathbf{M} - \mathbf{L}_\mathbf{M}) \right] \geq 0$.*

*Proof.* Note that

$$[(\mathbf{U_{IBP}} - \mathbf{L_{IBP}}) - (\mathbf{U_M} - \mathbf{L_M})] = \epsilon |\mathbf{a}_2^\top| |\mathbf{A}_1| \mathbf{1}_n + \frac{1}{2} |\mathbf{a}_2^\top| |\mathbf{u}_1| - \frac{1}{2} |\mathbf{a}_2^\top| |\mathbf{l}_1|$$
$$- 2\epsilon \left| |\mathbf{a}_2^\top \text{diag}\left( \mathbb{1}\{\mathbf{u}_1 \geq \mathbf{0}\} \right) \mathbf{A}_1 \right| \mathbf{1}_n$$

Consider the coordinate splitting functions $S^{++}(.), S^{+-}(.), S^{--}(.)$ and $S^{-+}(.)$ such that for $\mathbf{x} \in \mathbb{R}^n$ $S^{++}(\mathbf{x}) = \mathbf{x} \odot \mathbb{1}\{\mathbf{u}_1^i \geq 0, \mathbf{l}_1^i \geq 0\}$ where $\mathbb{1}\{\mathbf{u}_1^i \geq 0, \mathbf{l}_1^i \geq 0\}$ is a vector of all zeros and 1 in the locations where both $\mathbf{u}_1^i, \mathbf{l}_1^i \geq 0$. However, since $\mathbf{u}_1 \geq \mathbf{l}_1$, then $S^{-+}(.) = \mathbf{0}$. Therefore it is clear that for any vector $\mathbf{x}$ and an interval $[\mathbf{l}_1, \mathbf{u}_1]$, we have that

$$\mathbf{x} = S^{++}(\mathbf{x}) + S^{+-}(\mathbf{x}) + S^{--}(\mathbf{x}), \tag{9}$$

since the sets $\{i; \mathbf{u}_1^i \geq 0, \mathbf{l}_1^i \geq 0\}, \{i; \mathbf{u}_1^i \geq 0, \mathbf{l}_1^i \leq 0\}$ and $\{i; \mathbf{u}_1^i \leq 0, \mathbf{l}_1^i \leq 0\}$ are disjoints and their union $\{i = 1, i = 2, \ldots, i = k\}$. We will denote the difference in the interval lengths as $W_{IBP} - W_M$ for ease of notation. Thus, we have the following:

$$W_{IBP} - W_M = \epsilon S^{++}\left(|\mathbf{a}_2^\top|\right) |\mathbf{A}_1| \mathbf{1}_n + \epsilon S^{+-}\left(|\mathbf{a}_2^\top|\right) |\mathbf{A}_1| \mathbf{1}_n + \epsilon S^{--}\left(|\mathbf{a}_2^\top|\right) |\mathbf{A}_1| \mathbf{1}_n + \frac{1}{2} S^{++}\left(|\mathbf{a}_2^\top|\right) |\mathbf{u}_1|$$
$$+ \frac{1}{2} S^{+-}\left(|\mathbf{a}_2^\top|\right) |\mathbf{u}_1| + \frac{1}{2} S^{--}\left(|\mathbf{a}_2^\top|\right) |\mathbf{u}_1| - \frac{1}{2} S^{++}\left(|\mathbf{a}_2^\top|\right) |\mathbf{l}_1| - \frac{1}{2} S^{+-}\left(|\mathbf{a}_2^\top|\right) |\mathbf{l}_1|$$
$$- \frac{1}{2} S^{--}\left(|\mathbf{a}_2^\top|\right) |\mathbf{l}_1| - 2\epsilon \left| \left( S^{++}\left(\mathbf{a}_2^\top\right) + S^{+-}\left(\mathbf{a}_2^\top\right) + S^{--}\left(\mathbf{a}_2^\top\right) \right) \text{diag}\left( \mathbb{1}\{\mathbf{u}_1 \geq \mathbf{0}\} \right) \right| \mathbf{1}_n$$
$$= 2\epsilon S^{++}\left(|\mathbf{a}_2^\top|\right) |\mathbf{A}_1| \mathbf{1}_n + S^{+-}\left(|\mathbf{a}_2^\top|\right) (\mathbf{A}_1 \mathbf{x} + \mathbf{b}_1) + \epsilon S^{+-}\left(|\mathbf{a}_2^\top|\right) \mathbf{A}_1 \mathbf{1}_n$$
$$- 2\epsilon \left| \left( S^{++}\left(\mathbf{a}_2^\top\right) + S^{+-}\left(\mathbf{a}_2^\top\right) \right) \mathbf{A}_1 \right| \mathbf{1}_n$$
$$= \underbrace{2\epsilon S^{++}\left(|\mathbf{a}_2^\top|\right) |\mathbf{A}_1| \mathbf{1}_n}_{\text{①}} + \underbrace{S^{+-}\left(|\mathbf{a}_2^\top|\right) \mathbf{u}_1}_{\text{②}} - \underbrace{2\epsilon \left| \left( S^{++}\left(\mathbf{a}_2^\top\right) + S^{+-}\left(\mathbf{a}_2^\top\right) \right) \mathbf{A}_1 \right| \mathbf{1}_n}_{\text{③}}.$$

Note that we used the property of the coordinate splitting functions defined in Eq 9 along with the definitions of The previous The penultimate equality follows since $S^{++}(.)$ and $S^{+-}(.)$ corresponds to the indices that are selected by $\mathbf{l}_1$ and $\mathbf{u}_1$. The penultimate equality follows since $S^{++}$ and $S^{+-}$ corresponds to the indices that are selected by $\text{diag}\left( \mathbb{1}\{\mathbf{u}_1 \geq \mathbf{0}\} \right)$.

Now by taking the expectation over $\mathbf{a}_2$, we have for ① :

$$2\epsilon \mathbb{E}\left[ S^{++}\left(|\mathbf{a}_2^\top| |\mathbf{A}_1|\right) \right] \mathbf{1}_n = 2\epsilon \sum_{i=1}^k \mathbb{E}\left[ |\mathbf{a}_2^i| \right] |\mathbf{A}_1(i,:)| \mathbb{1}\{\mathbf{u}_1^i \geq \mathbf{0}, \mathbf{l}_1^i \geq \mathbf{0}\} \mathbf{1}_n$$
$$= 2\epsilon \sigma_{\mathbf{a}_2} \sqrt{\frac{2}{\pi}} \sum_{i=1}^k |\mathbf{A}_1(i,:)| \mathbb{1}\{\mathbf{l}_1^i \geq \mathbf{0}\} \mathbf{1}_n$$
$$= 2\epsilon \sigma_{\mathbf{a}_2} \sqrt{\frac{2}{\pi}} \sum_{j=1}^n \sum_{i=1}^k |\mathbf{A}_1(i,j)| \mathbb{1}\{\mathbf{l}_1^i \geq \mathbf{0}\}$$

The second equality follows from the mean of the folded Gaussian. and the fact that $\mathbf{u}_1 \geq \mathbf{l}_1$.

For ② , we have:

$$\mathbb{E}\left[S^{+-}\left(|\mathbf{a}_2^\top|\right)\mathbf{u}_1\right] = \sigma_{\mathbf{a}_2}\sqrt{\frac{2}{\pi}}\sum_{i=1}^k \mathbf{u}_1^i \mathbb{1}\left\{\mathbf{u}_1^i \geq 0, \mathbf{l}_1^i \leq 0\right\}$$

Lastly, for ③ , we have:

$$2\epsilon\mathbb{E}\left[\left\|\left(S^{++}\left(\mathbf{a}_2^\top\right) + S^{+-}\left(\mathbf{a}_2^\top\right)\right)\mathbf{A}_1\right\|\right]\mathbf{1}_n = 2\epsilon\mathbb{E}\left[\left\|\left[\sum_{i=1}^k \mathbf{A}_1(i,:)\mathbf{a}_2^i\left(\mathbb{1}\left\{\mathbf{u}_1^i \geq 0\right\}\right)\right]\right\|\right]\mathbf{1}_n$$

Using Holder's inequality, i.e. $\mathbb{E}[|x|] \leq \sqrt{\mathbb{E}[x^2]}$, per coordinate of the vector $\left[\sum_{i=1}^k \mathbf{A}_1(i,:)\mathbf{a}_2^i\left(\mathbb{1}\left\{\mathbf{u}_1^i \geq 0\right\}\right)\right]$ and by binomial expansion, we have at the $j^{\text{th}}$ coordinate

$$2\epsilon\sqrt{\mathbb{E}\left[\sum_{i=1}^k \mathbf{A}_1(i,j)\mathbf{a}_2^i\mathbb{1}\left\{\mathbf{u}_1^i \geq 0\right\}\right]^2}$$

$$= 2\epsilon\left(\sum_{i=1}^k (\mathbf{A}_1(i,j))^2\,\mathbb{E}\left[\left(\mathbf{a}_2^i\right)^2\right]\mathbb{1}\left\{\mathbf{u}_1^i \geq 0\right\} + 2\sum_{i=1}\sum_{z<i}\mathbf{A}_1(i,j)\mathbf{A}_1(z,j)\mathbb{E}\left[\mathbf{a}_2^i\mathbf{a}_2^z\right]\mathbb{1}\left\{\mathbf{u}_1^i \geq 0\right\}\mathbb{1}\left\{\mathbf{u}_1^z \geq 0\right\}\right)^{\frac{1}{2}}$$

$$= 2\epsilon\sqrt{\sum_{i=1}^k (\mathbf{A}_1(i,j))^2\,\mathbb{E}\left[\left(\mathbf{a}_2^i\right)^2\right]\mathbb{1}\left\{\mathbf{u}_1^i \geq 0\right\}} = 2\epsilon\sigma_{\mathbf{a}_2}\sqrt{\sum_{i=1}^k (\mathbf{A}(i,j))^2\,\mathbb{1}\left\{\mathbf{u}_i \geq 0\right\}}$$

The second equality follows from by the independence of $\mathbf{a}_2^i$ and that they have zero mean. Therefore it follows from ③ that:

$$2\epsilon\mathbb{E}\left[\left\|\left(S^{++}\left(\mathbf{A}_2^\top\right) + S^{+-}\left(\mathbf{A}_2^\top\right)\right)\mathbf{A}_1\right\|\right]\mathbf{1}_n \leq 2\epsilon\sigma_{\mathbf{a}_2}\sum_{j=1}^n\sqrt{\sum_{i=1}^k (\mathbf{A}_1(i,j))^2\,\mathbb{1}\left\{\mathbf{u}_i \geq 0\right\}}$$

Lastly, putting things together, i.e. $\mathbb{E}\left[① + ② - ③\right]$ we have that

$$\mathbb{E}\left[W_{IBP} - W_M\right] \geq 2\epsilon\sigma_{\mathbf{a}_2}\sqrt{\frac{2}{\pi}}\sum_{j=1}^n\sum_{i=1}^k |\mathbf{A}_1(i,j)|\mathbb{1}\left\{\mathbf{l}_1^i \geq 0\right\} + \sigma_{\mathbf{a}_2}\sqrt{\frac{2}{\pi}}\sum_{i=1}^k \mathbf{u}_1^i\mathbb{1}\left\{\mathbf{u}_1^i \geq 0, \mathbf{l}_1^i \leq 0\right\}$$

$$- 2\epsilon\sigma_{\mathbf{a}_2}\sum_{j=1}^n\sqrt{\sum_{i=1}^k \mathbf{A}_1(i,j)^2\,\mathbb{1}\left\{\mathbf{u}_1^i \geq 0\right\}}.$$

(10)

Note that to show that the previous inequality is non-negative, it is sufficient to show that the previous inequality is non-negative for the non-intersecting sets $\{i : \mathbf{l}_1^i \geq 0\}$ and $\{i : \mathbf{u}_1^i \geq 0, \mathbf{l}_1^i \leq 0\}$. Thus the right hand side can be written as the sum of two sets.

**For the set** $\{i : \mathbf{l}_1^i \geq 0\}$, the RHS of inequality 10 reduces to

$$2\epsilon\sigma_{\mathbf{a}_2}\sum_{j=1}^n\left(\sqrt{\frac{2}{\pi}}\|\mathbf{A}_1(:,j)\|_1 - \|\mathbf{A}_1(:,j)\|_2\right).$$

(11)

**For the set** $\{i : \mathbf{u}_1^i \geq 0, \mathbf{l}_1^i \leq 0\}$ and using the definition of $\mathbf{u}_1$, the RHS of inequality 10 reduces to

$$\sigma_{\mathbf{a}_2}\sqrt{\frac{2}{\pi}}\sum_{i=1}^{k}\left(\sum_{j=1}^{n}\mathbf{A}_1(i,j)\mathbf{x}_j + \mathbf{b}_i + \epsilon\sum_{j=1}^{n}|\mathbf{A}_1(i,j)|\right) - 2\epsilon\sigma_{\mathbf{a}_2}\sum_{j=1}^{n}\|\mathbf{A}_1(:,j)\|_2$$

$$= \sigma_{\mathbf{a}_2}\sqrt{\frac{2}{\pi}}\sum_{j=1}^{n}\left(\mathbf{x}_j\mathbf{1}_k^\top\mathbf{A}_1(:,j) + \frac{1}{n}\mathbf{1}_k^\top\mathbf{b} + \epsilon\|\mathbf{A}_1(:,j)\|_1\right) - 2\epsilon\sigma_{\mathbf{a}_2}\sum_{j}\|\mathbf{A}_1(:,j)\|_2$$

$$= \sum_{j=1}^{n}\left(\sigma_{\mathbf{a}_2}\sqrt{\frac{2}{\pi}}\left(\mathbf{x}_j\mathbf{1}_k^\top\mathbf{A}_1(:,j) + \frac{1}{n}\mathbf{1}_k^\top\mathbf{b} + \epsilon\|\mathbf{A}_1(:,j)\|_1\right) - 2\epsilon\sigma_{\mathbf{a}_2}\|\mathbf{A}_1(:,j)\|_2\right) \qquad (12)$$

Note that given the assumption in the Theorem where $\frac{1}{\sqrt{2\pi}}\mathbf{x}_j\mathbf{1}_k^\top\mathbf{A}_1(:,j) + \frac{1}{2n}\mathbf{1}_k^\top\mathbf{b} \geq 0 \geq \epsilon\left(\|\mathbf{A}_1(:,j)\|_2 - \frac{1}{\sqrt{2\pi}}\|\mathbf{A}_1(:,j)\|_1\right) \forall j$, then if both Eq 11 and Eq 12 are non-negative completing the proof. $\qquad\square$

**Lemma 1.** *For $\mathbf{x} \in \mathbb{R}^k \sim \mathcal{N}\left(\mathbf{0},\mathbf{I}\right)$, where $k \geq 5$ we have that $\mathbb{E}\left[\frac{3}{\sqrt{2\pi}}\|\mathbf{x}\|_1 - 2\|\mathbf{x}\|_2\right] \geq 0$.*

*Proof.* Note that by the mean of a folded Gaussian, we gave that $\mathbb{E}\left[\|\mathbf{x}\|_1\right] = \sum_i^k \mathbb{E}\left[|\mathbf{x}_i|\right] = k\sqrt{\frac{2}{\pi}}$. Moreover, note that

$$\mathbb{E}\left[\|\mathbf{x}\|_2\right] = \mathbb{E}\left[\sqrt{\sum_i^k\mathbf{x}_i^2}\right] = \mathbb{E}\left[\sqrt{y}\right] = \frac{1}{2^{\frac{k}{2}-1}\Gamma(\frac{k}{2})}\int_0^\infty x^k\exp\left(-\frac{x^2}{2}\right)dx = \frac{2^{\frac{k-1}{2}}\Gamma\left(\frac{k+1}{2}\right)}{2^{\frac{k}{2}-1}\Gamma(\frac{k}{2})}$$

$$= \sqrt{2}\frac{\Gamma\left(\frac{k+1}{2}\right)}{\Gamma(\frac{k}{2})} \sim \sqrt{k}.$$

Note that $y$ is Chi-Square random variable and that $f_{\sqrt{y}}(x) = 2x f_y(x^2) = \frac{x^{k-1}}{2^{\frac{k}{2}-1}\Gamma(\frac{k}{2})}\exp\left(-\frac{x^2}{2}\right)$ where the third inequality follows by integrating by parts recursively. Lastly, the last approximation follows by stirling's approximation for large $k$. $\qquad\square$

**Proposition 3.** *For a random matrix $\mathbf{A}_1 \in \mathbb{R}^{k\times n}$ with i.i.d elements such $\mathbf{A}_1(i,j) \sim \mathcal{N}(0,1)$, then*

$$\mathbb{E}_{\mathbf{A}_1}\left(\|\mathbf{A}_1(:,j)\|_2 - \frac{1}{\sqrt{2\pi}}\|\mathbf{A}_1(:,j)\|_1\right) = \sqrt{2}\frac{\Gamma\left(\frac{k+1}{2}\right)}{\Gamma\left(\frac{k}{2}\right)} - k\sqrt{\frac{2}{\pi}} \approx \sqrt{k}\left(1 - \sqrt{\frac{2}{\pi}}\sqrt{k}\right).$$

*Proof.* The proof follows immediately from Lemma 1. $\qquad\square$

## A   MORE QUALITATIVE RESULTS OF THE NEW BOUNDS

We conduct several more experiments to showcase the tightness of our proposed bounds to the true bounds and compared them against propagating the bounds layerwise $[\mathbf{l}_{\text{DM}}, \mathbf{u}_{\text{DM}}]$ for random $n - 100 - 100 - 100 - 100 - 2$ networks initialized with $\mathcal{N}(0, 1/\sqrt{n})$ similar to Wong & Kolter (2018). We show our bounds compared to the polytobe estimated from MonteCarlo sampling on results for $n \in \{2, 10, 20\}$ and $\epsilon \in \{0.05, 0.1, 0.25\}$. The layer wise bound propagation is shown in the tables as the bounds were too loose to be presented visually.

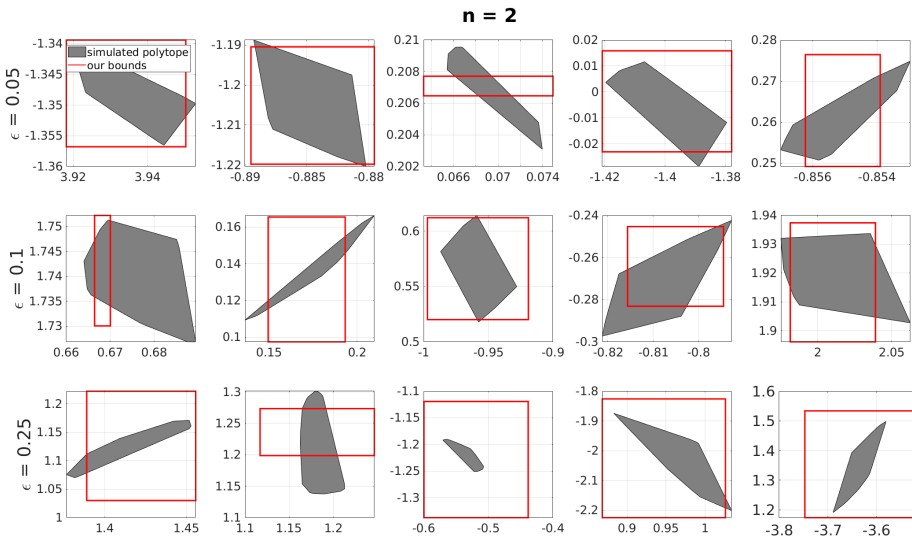

Figure 7: Each row represents 5 different randomly initialized networks for a given $\epsilon$ with $n = 2$. Note that the proposed bounds are far from being true this is as predicted by Theorem 1 for small $n$.

| $\epsilon$, Figure Number | $l_{\text{DM}}^1$ | $u_{\text{DM}}^1$ | $l_{\text{DM}}^2$ | $u_{\text{DM}}^2$ |
|---|---|---|---|---|
| $\epsilon = 0.05$, Figure Number = 1 | -10.1261 | 19.0773 | -18.1500 | 13.3573 |
| $\epsilon = 0.05$, Figure Number = 2 | -12.2529 | 14.3428 | -14.4295 | 12.3479 |
| $\epsilon = 0.05$, Figure Number = 3 | -12.6594 | 14.1837 | -12.5873 | 12.2612 |
| $\epsilon = 0.05$, Figure Number = 4 | -17.7825 | 16.4048 | -15.3843 | 15.1688 |
| $\epsilon = 0.05$, Figure Number = 5 | -12.5260 | 11.1149 | -8.9242 | 12.7539 |
| $\epsilon = 0.1$, Figure Number = 1 | -27.4598 | 23.6603 | -17.9481 | 23.4817 |
| $\epsilon = 0.1$, Figure Number = 2 | -23.2877 | 34.0542 | -28.1535 | 21.8703 |
| $\epsilon = 0.1$, Figure Number = 3 | -35.2950 | 36.4901 | -31.7465 | 36.0421 |
| $\epsilon = 0.1$, Figure Number = 4 | -31.7154 | 29.3062 | -30.3900 | 35.7105 |
| $\epsilon = 0.1$, Figure Number = 5 | -25.0870 | 39.4373 | -24.5087 | 32.5493 |
| $\epsilon = 0.25$, Figure Number = 1 | -54.0557 | 56.2884 | -52.5686 | 73.9621 |
| $\epsilon = 0.25$, Figure Number = 2 | -59.2115 | 82.8742 | -75.7999 | 65.6898 |
| $\epsilon = 0.25$, Figure Number = 3 | -50.1142 | 56.2330 | -72.4221 | 54.4631 |
| $\epsilon = 0.25$, Figure Number = 4 | -52.6030 | 83.3950 | -92.8100 | 69.1401 |
| $\epsilon = 0.25$, Figure Number = 5 | -89.1335 | 43.4685 | -74.4519 | 91.5137 |

Table 2: Shows the interval bounds obtained by propagating $\epsilon \in \{0.05, 0.1, 0.25\}$ with $n = 2$ and denoted as $l_{\text{DM}}^1, u_{\text{DM}}^1$ for the first output function of the 2-dimensional output network (shown along the x-axis in the previous figure) while $l_{\text{DM}}^2$ and $u_{\text{DM}}^2$ is for the other function (shown along the y-axis in the previous figure).

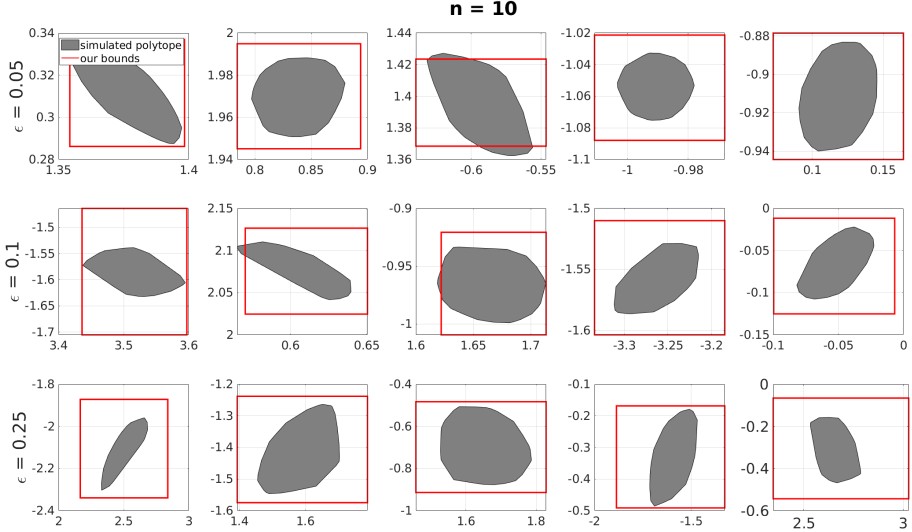

Figure 8: Each row represents 5 different randomly initialized networks for a given $\epsilon$ with $n = 10$. Note how the bounds are more likely now to enclose the true output region for all given $\epsilon$ compared to previous case where $n = 2$.

| $\epsilon$, Figure Number | $l_{\mathrm{DM}}^1$ | $u_{\mathrm{DM}}^1$ | $l_{\mathrm{DM}}^2$ | $u_{\mathrm{DM}}^2$ |
|---|---|---|---|---|
| $\epsilon = 0.05$, Figure Number $= 1$ | -16.9716 | 24.9259 | -21.2584 | 20.6358 |
| $\epsilon = 0.05$, Figure Number $= 2$ | -42.6267 | 48.1786 | -38.6958 | 37.7851 |
| $\epsilon = 0.05$, Figure Number $= 3$ | -41.4147 | 36.4056 | -42.0363 | 36.6605 |
| $\epsilon = 0.05$, Figure Number $= 4$ | -32.1013 | 25.1485 | -37.7864 | 33.3652 |
| $\epsilon = 0.05$, Figure Number $= 5$ | -45.4368 | 32.9774 | -44.8946 | 38.6805 |
| $\epsilon = 0.1$, Figure Number $= 1$ | -48.1221 | 86.6800 | -54.3059 | 71.2724 |
| $\epsilon = 0.1$, Figure Number $= 2$ | -51.2668 | 46.1237 | -38.8089 | 33.6512 |
| $\epsilon = 0.1$, Figure Number $= 3$ | -51.3915 | 52.4437 | -52.7149 | 49.1031 |
| $\epsilon = 0.1$, Figure Number $= 4$ | -71.7738 | 54.4836 | -91.0335 | 37.0950 |
| $\epsilon = 0.1$, Figure Number $= 5$ | -48.1744 | 33.2927 | -40.9540 | 47.2282 |
| $\epsilon = 0.25$, Figure Number $= 1$ | -152.7639 | 192.4156 | -188.4030 | 148.2482 |
| $\epsilon = 0.25$, Figure Number $= 2$ | -196.8923 | 195.4355 | -163.2691 | 177.3766 |
| $\epsilon = 0.25$, Figure Number $= 3$ | -141.6800 | 207.5414 | -207.9396 | 190.2823 |
| $\epsilon = 0.25$, Figure Number $= 4$ | -200.7513 | 156.2560 | -227.6427 | 182.0180 |
| $\epsilon = 0.25$, Figure Number $= 5$ | -153.3898 | 164.8314 | -147.8662 | 137.4380 |

Table 3: Shows the interval bounds obtained by propagating $\epsilon \in \{0.05, 0.1, 0.25\}$ with $n = 10$ and denoted as $l_{\mathrm{DM}}^1$, $u_{\mathrm{DM}}^1$ for the first output function of the 2-dimensional output network (shown along the x-axis in the previous figure) while $l_{\mathrm{DM}}^2$ and $u_{\mathrm{DM}}^2$ is for the other function (shown along the y-axis in the previous figure).

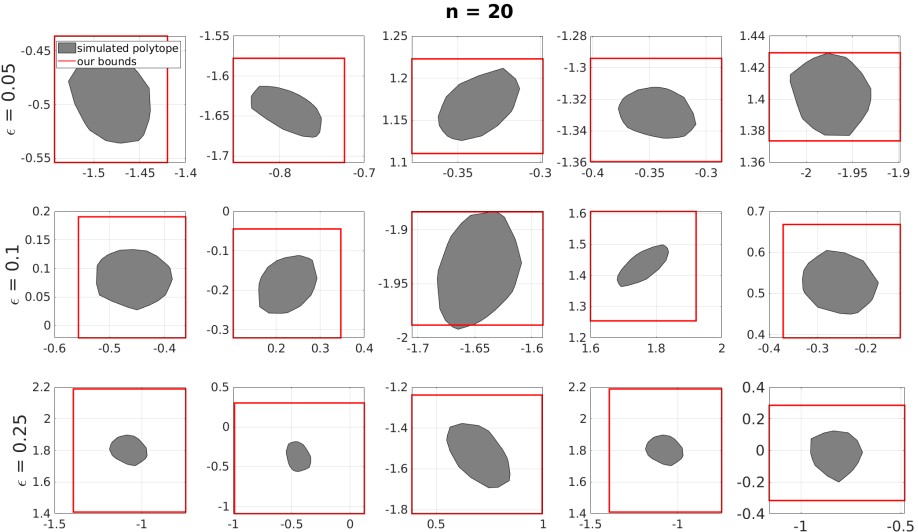

Figure 9: Each row represents 5 different randomly initialized networks for a given $\epsilon$ with $n = 20$. The bounds almost always enclose the polytope computed from Monte-Carlo compared to $n = 2, 10$.

| $\epsilon$, Figure Number | $l_{\text{DM}}^1$ | $u_{\text{DM}}^1$ | $l_{\text{DM}}^2$ | $u_{\text{DM}}^2$ |
|---|---|---|---|---|
| $\epsilon = 0.05$, Figure Number $= 1$ | -43.6689 | 32.8572 | -47.7856 | 36.9842 |
| $\epsilon = 0.05$, Figure Number $= 2$ | -53.2447 | 47.1651 | -46.5306 | 53.1638 |
| $\epsilon = 0.05$, Figure Number $= 3$ | -59.1694 | 42.6647 | -43.4659 | 57.2781 |
| $\epsilon = 0.05$, Figure Number $= 4$ | -39.8479 | 42.4197 | -42.1962 | 39.7649 |
| $\epsilon = 0.05$, Figure Number $= 5$ | -54.3150 | 42.8637 | -44.5742 | 43.8117 |
| $\epsilon = 0.1$, Figure Number $= 1$ | -83.5804 | 81.9034 | -97.5203 | 98.8713 |
| $\epsilon = 0.1$, Figure Number $= 2$ | -64.8464 | 76.8083 | -84.9223 | 83.9505 |
| $\epsilon = 0.1$, Figure Number $= 3$ | -70.5862 | 92.6652 | -88.6098 | 71.8915 |
| $\epsilon = 0.1$, Figure Number $= 4$ | -78.0557 | 151.4360 | -106.073 | 123.3686 |
| $\epsilon = 0.1$, Figure Number $= 5$ | -91.8368 | 97.3438 | -103.2845 | 76.6581 |
| $\epsilon = 0.25$, Figure Number $= 1$ | -188.7623 | 256.2275 | -211.3972 | 255.5101 |
| $\epsilon = 0.25$, Figure Number $= 2$ | -219.5642 | 274.5287 | -217.7622 | 349.4256 |
| $\epsilon = 0.25$, Figure Number $= 3$ | -214.7457 | 160.7498 | -186.5554 | 184.1767 |
| $\epsilon = 0.25$, Figure Number $= 4$ | -188.7623 | 256.2275 | -211.3972 | 255.5101 |
| $\epsilon = 0.25$, Figure Number $= 5$ | -276.9137 | 177.7929 | -202.2031 | 245.8731 |

Table 4: Shows the interval bounds obtained by propagating $\epsilon \in \{0.05, 0.1, 0.25\}$ with $n = 20$ and denoted as $l_{\text{DM}}^1$, $u_{\text{DM}}^1$ for the first output function of the 2-dimensional output network (shown along the x-axis in the previous figure) while $l_{\text{DM}}^2$ and $u_{\text{DM}}^2$ is for the other function (shown along the y-axis in the previous figure).

## B    Experimental Setup for Training DNNs

| small | medium | large |
|---|---|---|
| CONV $16 \times 4 \times 4 + 2$ | CONV $32 \times 3 \times 3 + 1$ | CONV $64 \times 3 \times 3 + 1$ |
| CONV $32 \times 4 \times 4 + 1$ | CONV $32 \times 4 \times 4 + 2$ | CONV $64 \times 3 \times 3 + 1$ |
| FC 100 | CONV $64 \times 3 \times 3 + 1$ | CONV $128 \times 3 \times 3 + 2$ |
|  | CONV $64 \times 4 \times 4 + 2$ | CONV $128 \times 3 \times 3 + 1$ |
|  | FC 512 | CONV $128 \times 3 \times 3 + 1$ |
|  | FC 512 | FC 200 |

Table 5: Architectures for the three models trained on MNIST and CIFAR10. "CONV $p \times w \times h + s$", correspond to $p$ 2D convolutional filters with size $(w \times h)$ and strides of $s$. While "FC $d$" is a fully connected layer with d outputs. Note that the last fully connected layer is omitted.

# C PGD ROBUSTNESS ON SPECIFIC INPUT BOUNDS ON MNIST

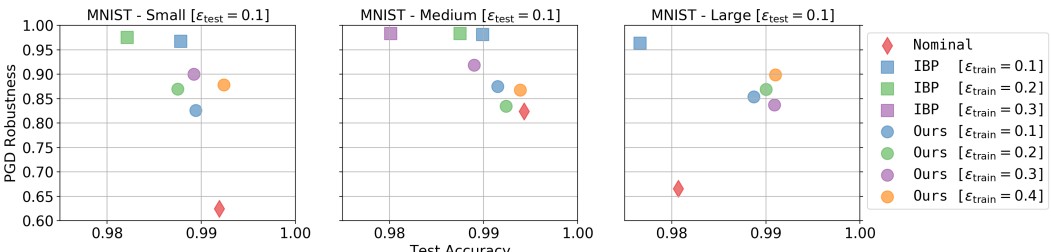

Figure 10: Compares PGD ($\epsilon_{\text{test}} = 0.1$) and test accuracy of our models against IBP on MNIST.

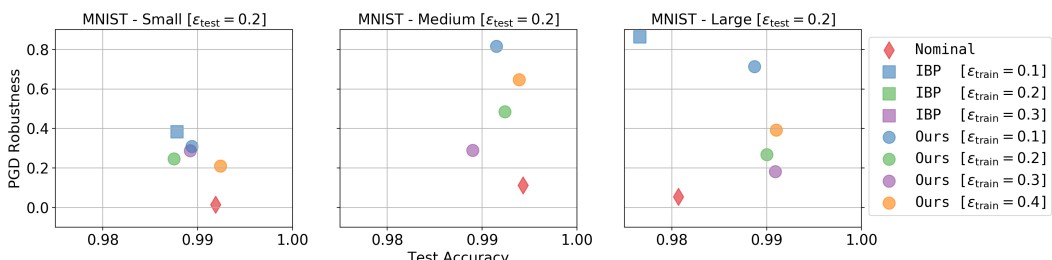

Figure 11: Compares PGD ($\epsilon_{\text{test}} = 0.2$) and test accuracy of our models against IBP on MNIST.

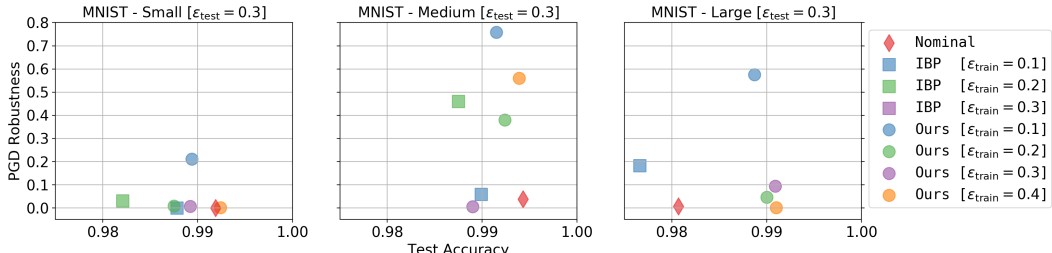

Figure 12: Compares PGD ($\epsilon_{\text{test}} = 0.3$) and test accuracy of our models against IBP on MNIST.

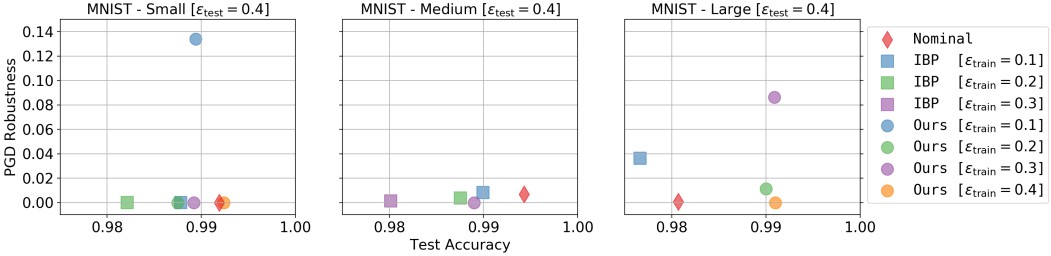

Figure 13: Compares PGD ($\epsilon_{\text{test}} = 0.4$) and test accuracy of our models against IBP on MNIST.

# D  PGD ROBUSTNESS ON SPECIFIC INPUT BOUNDS ON CIFAR10

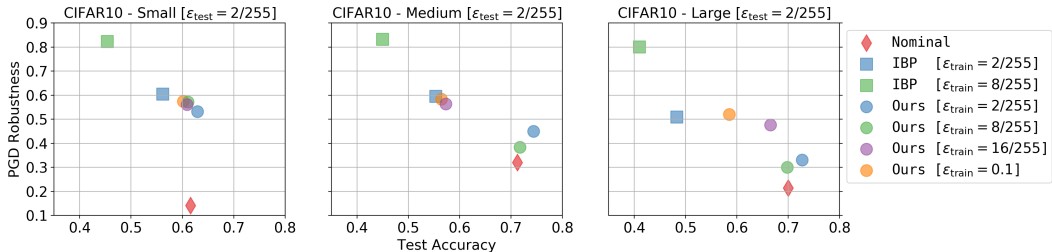

Figure 14: Compares PGD ($\epsilon_{\text{test}} = {}^2\!/_{255}$) and test accuracy of our models against IBP on CIFAR10.

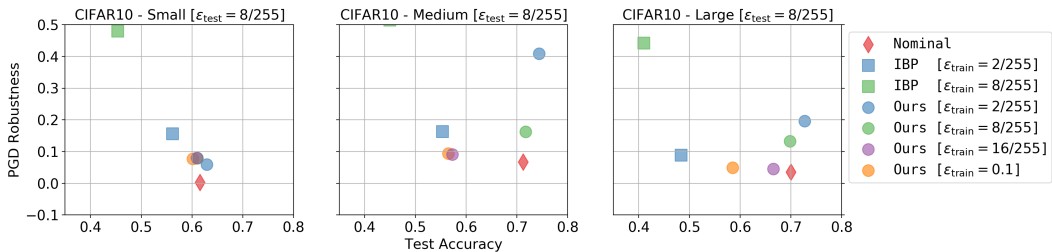

Figure 15: Compares PGD ($\epsilon_{\text{test}} = {}^8\!/_{255}$) and test accuracy of our models against IBP on CIFAR10.

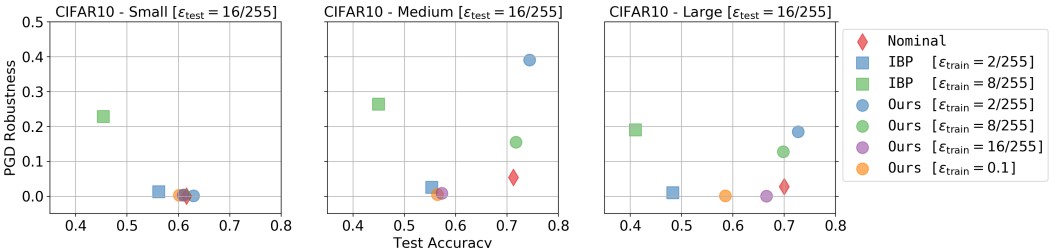

Figure 16: Compares PGD ($\epsilon_{\text{test}} = {}^{16}\!/_{255}$) and test accuracy of our models against IBP on CIFAR10.

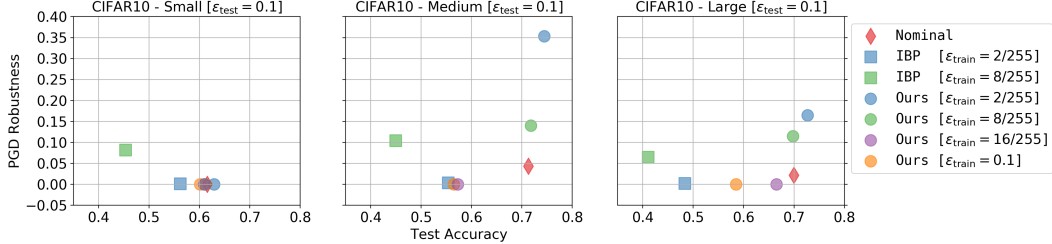

Figure 17: Compares PGD ($\epsilon_{\text{test}} = 0.1$) and test accuracy of our models against IBP on CIFAR10.

## E  FAILURE EXAMPLE FOR OUR BOUNDS

In this section, we show a faliure example to when the network weights do not follow Gaussian Assumption 1, our bounds can be far too loose even compared to IBP. Consider a two layer neural network where $\mathbf{A}_1 = 1000\, \mathbf{I}_{n\times n}$, $\mathbf{b}_1 = -999\, \mathbf{1}_n$, $\mathbf{a}_2 = -10\, \mathbf{1}_n$ and $b_2 = 0$ for the interval $[-\mathbf{1}_n, \mathbf{1}_n]$. Then, we have that

$$\mathbf{l}, \mathbf{u} = \mathbf{b}_1 \mp \epsilon |\mathbf{A}_1| \mathbf{1}_n$$
$$= \mathbf{1}_n \left( \mp 1000 - 999 \right) = -1999\, \mathbf{1}_n, \mathbf{1}_n.$$

Since $\mathbf{u} \geq \mathbf{0}$, then $\mathbf{M} = \mathbf{I}_{n\times n}$. Thereafter our estimated bounds as given by Equation 3 are given as:

$$\mathbf{L_M}, \mathbf{U_M} = \mathbf{a}_2^\top \mathbf{b}_1 \mp \epsilon |\mathbf{a}_2^\top \mathbf{A}_1| \mathbf{1}_n$$
$$= 9990n \mp 10000\, \epsilon \mathbf{1}_n^\top \mathbf{1}_n = -10n, 19990n.$$

As for IBP bounds, they are given as follows:

$$\mathbf{L_{IBP}}, \mathbf{U_{IBP}} = \mathbf{a}_2^\top \left( \frac{\max(\mathbf{u}, \mathbf{0}_n) + \max(\mathbf{l}, \mathbf{0})}{2} \right) \mp |\mathbf{a}_2^\top| \left( \frac{\max(\mathbf{u}, \mathbf{0}_n) - \max(\mathbf{l}, \mathbf{0})}{2} \right)$$
$$= -\frac{10}{2} \mathbf{1}_n^\top \mathbf{1}_n \mp \frac{10}{2} \mathbf{1}_n^\top \mathbf{1}_n = -10n, 0.$$

Under this construction of weights to the network, different of the i.i.d Gaussian assumption, it is clear that our bounds can be orders of magnitude looser to IBP. In the following section, we demonstrate that networks trained on real data do have weights that are not far from the Gaussian assumption by empirically investigating the histogram of the network weights.

# F HISTOGRAM OF WEIGHTS OF TRAINED NETWORKS

To quantify how reasonable is the assumption of Gaussian i.i.d weights in trained networks, we train medium CNN (discussed in previous section) and show the histogram of the weights of the first 6 layers. The network is trained with and without $\ell_2$ regularization on three datasets namely MNIST, CIFAR10 and CIFAR100.

## F.1 HISTOGRAM OF WEIGHTS ON MNIST

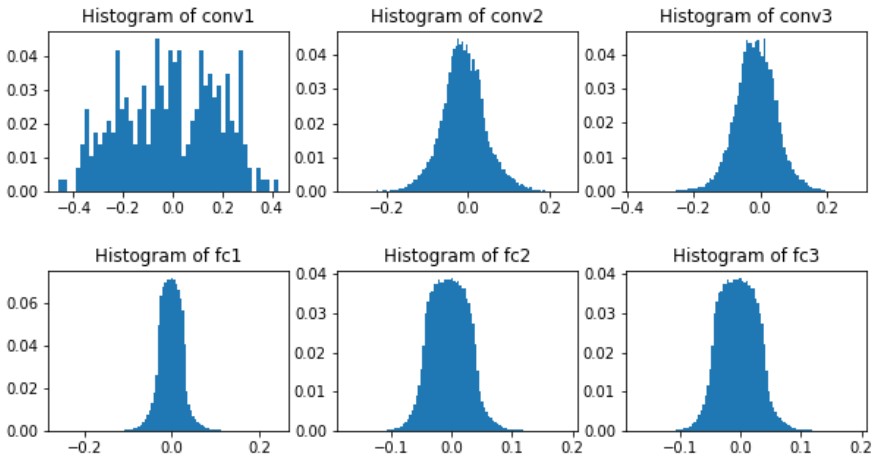

Figure 18: Histogram of weights of medium CNN trained on MNIST without $\ell_2$ regularization.

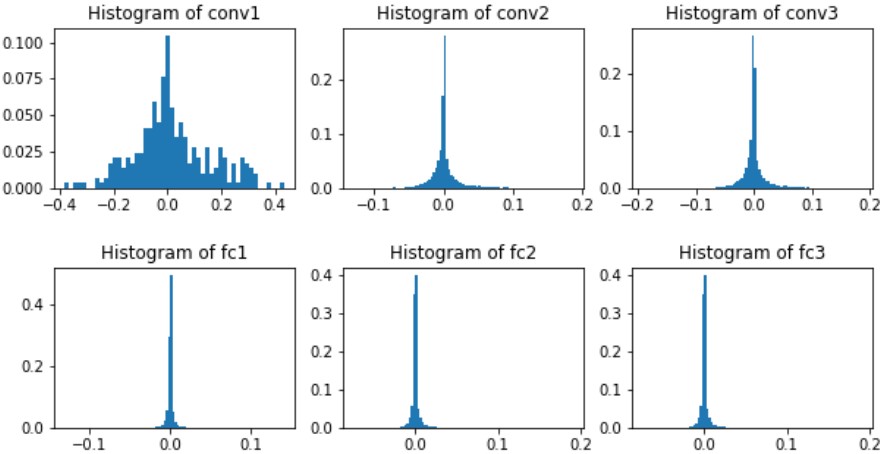

Figure 19: Histogram of weights of medium CNN trained on MNIST with $\ell_2$ regularization.

## F.2 HISTOGRAM OF WEIGHTS ON CIFAR10

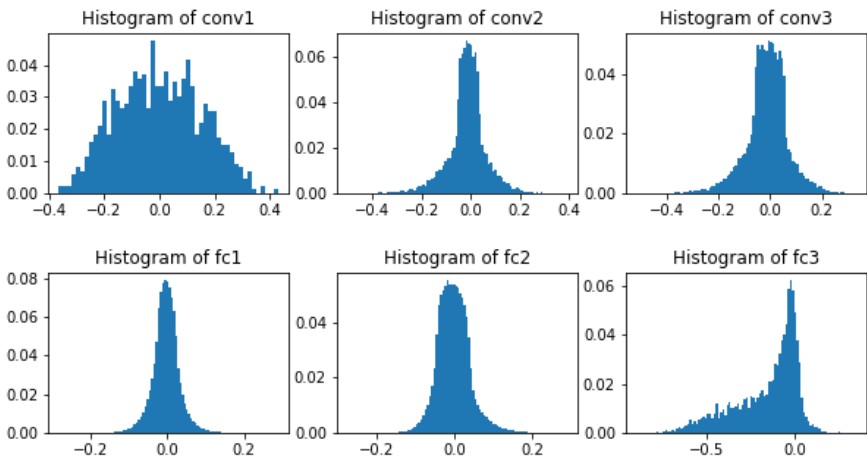

Figure 20: Histogram of weights of medium CNN trained on CIFAR10 without $\ell_2$ regularization.

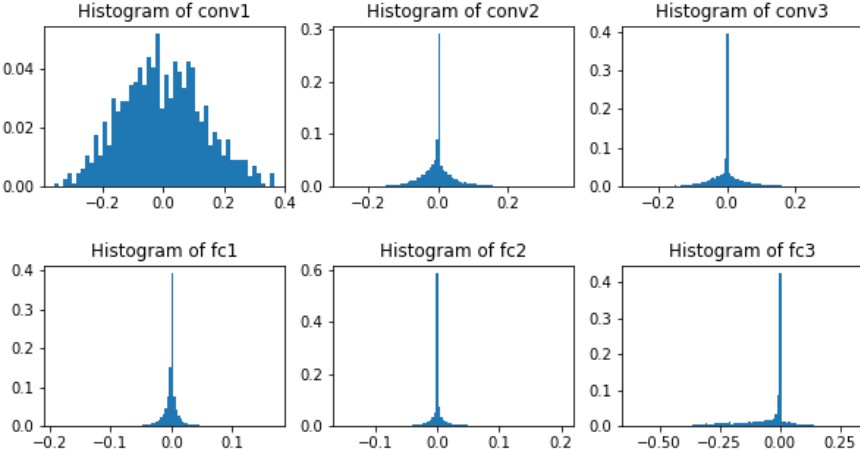

Figure 21: Histogram of weights of medium CNN trained on CIFAR10 with $\ell_2$ regularization.

### F.3  HISTOGRAM OF WEIGHTS ON CIFAR100

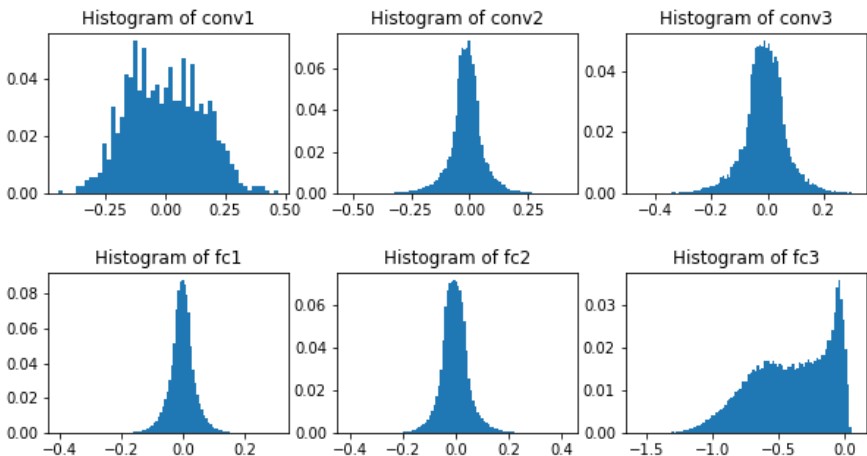

Figure 22: Histogram of weights of medium CNN trained on CIFAR100 without $\ell_2$ regularization.

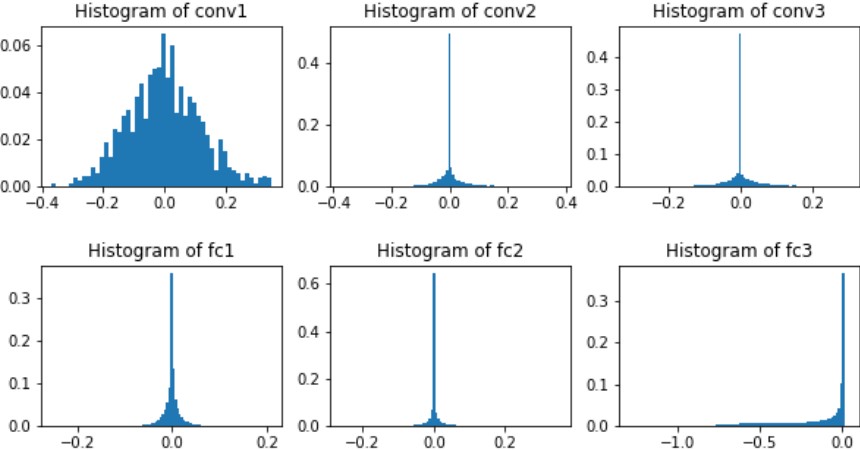

Figure 23: Histogram of weights of medium CNN trained on CIFAR100 with $\ell_2$ regularization.

# G    Rebuttal

## G.1    Reviewer 2

We thank R2 for their very thorough review of our paper. Regarding the disclaimer, we want to clarify to other reviewers too that we have indeed tried to address all R2's previous concerns in this submission. We have conducted some extra experiments from the previous submission; we have also done some serious changes to the structure and presentation of the paper as previously suggested by R2. In what follows we try addressing R2's current concerns.

• **The logical argument explaining the "why" these are valid is harder to understand and could be clarified. My understanding so far ...**

The overall description of R2 to the logical argument is correct. The proof is based on showing that our bounds are 'supersets' to some approximate bounds $\mathbf{L}_{\text{approx}}$ and $\mathbf{U}_{\text{approx}}$. The approximate bounds are indeed related to the unknown true bounds through Assumption 1. Note that Assumption 1 states that there exists some $m$ where the Assumption inequality holds. As we pointed out to R1, the larger the constant $m$, the larger $n$ has to be for Theorem 1 to hold. Please refer to the last inequality in the proof of Theorem 1.

• **The caption of Figure 1 is misleading. It says that it shows that the proposed ...**

True. This was a mistake of phrasing on our part. We have addressed this in the revised version.

• **The reporting of Figure 5 and 6 is weird because according to the text, each datapoint seems to be the average robustness of ...**

We have addressed this in the revised version. The reported models had only their robustness averaged over multiple $\epsilon_{\text{test}}$. Please note that the robustness results over each individual $\epsilon_{\text{test}}$ were left for the appendix. The best models from the grid search over the hyperparameters are reported for both our bound training and IBP training for fair comparison.

• **I appreciate the effort of the author to include experiments involving a MIP solver returning the true bounds. This is very helpful in building confidence that the bounds generated are correct. How is the real network trained? Is this based on a robustly trained network or is it just standard training? Is 99% the nominal accuracy?**

Yes, $99\%$ is the nominal accuracy of the trained models. The network was trained normally without any special regularization.

• **In general, I think that the paper would be better if there was more discussion of the failure modes of the method. There are easy to identify failure cases where the proposed bound is incorrect or loose. My opinion is that the paper would be stronger if it acknowledged them ...**

We agree with R2. We have addressed this in the revised version. In particular, we reflected this in the paragraph just before section 3.3. We have added in the appendix the example of the failure case pointed out by R2. Moreover, as suggested by R1, we have investigated, at least empirically, the histogram of the weights of a normally trained network on 3 real dataset in appendix F. In what follows, we re-state our response to R1 in regards to the histogram results of the appendix to R2 for completion. "For the MNIST experiments, even without $\ell_2$ regularization, the weights seem to follow a bell shaped. With $\ell_2$ regularization, the histogram tend to be much pointy. Of course, the regularization parameter used in training interpolates between the two histograms, the larger the regularization the more 'pointy' the histogram will tend to be. A similar observation can be noted for CIFAR10 and CIFAR100. We want to emphasize that while this is not a proof that the Gaussian assumption holds in practice, we believe that at least the distribution of the weights does not seem to be far from Gaussian and thereafter sheds light on why perhaps such an assumption may not be utterly unreasonable on real networks."

• **The assumption about gaussian weights in A1 and a2 seems strong but is at least partially motivated at the end of 3.2 (although not all networks are ...**

R2 is correct. The i.i.d assumption over all elements of the weights matrix $\mathbf{A}$ breaks in the presence of topelitz/circulant structure convolutional layers have. This, unfortunately, is as far as the current theory can go. We find though that this have little impact on practically upon training the networks.

• **I think that the paper would benefit from having some discussion of other methods that can derive "bounds" which may not actually be bounds. The works by Stefan Webb (A Statistical ...**

Indeed, the two referenced works are related to our direction since both aim at either probabilistically carrying out the verification or by estimating the lower bound of the minimum perturbation. We elaborated on this in the revised version and added a paragraph in the introduction.

• **While reviewing the paper, I also spotted some strong similarities between the methods proposed (particularly subsection 3.3) and the Fastlin method. The computation mechanism of Fastlin (propagate recursively from the end through the linear layers and through diagonal matrices that replace the ReLU activation function) is exactly the same as the proposed one (Fastlin goes through the ...**

The description of R1 for the intuition of the construction of the diagonal matrix $\mathbf{M}$ is correct. Indeed, there are similarities between both our work and Fastlin. However, we do not believe that the works are limited. Both works still provide an excellent trade-off between efficiency and computational complexity, even when over estimating the intermediate upper bounds for the intermediate $\mathbf{M}$, the approximation for the superset bounds are still effective.

• **Minor comments, typos and references.** We have addressed the issues in the revised version. We have also tried to address other issues and writing style highlighted by R2.

## G.2 REVIEWER 1

We thank R1 for the time spent in reviewing our paper. Follows our response to R1's concerns.

• **Some assumptions seem pretty unrealistic to me. For instance, the fact that the neural network should have Gaussian i.i.d weights: it is thus ...**

As correctly pointed out by R1, Gowal et al do not make such an assumption. They provide loose 'true' bounds for deep architectures as a function of network parameters by simply computing worst case bound per layer. As suggested by R1, we have conducted an experiment where we trained Medium CNN with identical structure to previous experiments on MNIST, CIFAR10 and on CIFAR100 with and without $\ell_2$ regularization. We then plot the histogram of the weights of the 6 layers in the network (3 convolutional and 3 fully connected). We report the results in Section E of the appendix in the revised version. For the MNIST experiments, even without $\ell_2$ regularization, the weights seem to follow a bell shaped. With $\ell_2$ regularization, the histogram tend to be much pointy. Of course, the regularization parameter used in training interpolates between the two histograms, the larger the regularization the more 'pointy' the histogram will tend to be. A similar observation can be noted for CIFAR10 and CIFAR100. We want to emphasize that while this is not a proof that the Gaussian assumption holds in practice, we believe that at least the distribution of the weights does not seem to be far from Gaussian and thereafter sheds light on why perhaps such an assumption may not be utterly unreasonable on real networks.

• **A major difference with (Gowal et al) is that: here, the bounds are in expectation whereas the bound in (Gowal et al) are deterministic. In this paper, there are some approximation assumptions ...**

This is true. Unfortunately, as far as the analysis goes, we do not yet have concentration results. One direction of interest is to study the asymptotic behaviour (in terms of $n$ and $k$) of the variance of $\mathbf{L_M}$ and $\mathbf{U_M}$ over the distribution of network parameters. So far we have some little progress in that direction as the analysis becomes untraceable.

• **Most of the proofs are long and complicated, and several equations of 7-8 lines could be summarized with up to 2-3 lines maximum. I had a hard time to understand the proof of Theorem 1, which is simply some algebra. This could be improved.**

The equations were, for most part purposely, left to span multiple lines as we wanted to elaborate on each step of the analysis.

• **I'm curious of the imagenet performance: couldn't this technique be easily applied to AlexNet, which is nowadays simple to manipulate? Is there a technical issue to do so?**

While we were interested for the most part in the analysis, we do believe that the paper can benefit from the ImageNet experiments vastly. They were left out for time/computation capacity related reasons. We will try to address this within our capacity.

• **Is the assumption 1 really an assumption? given a,a' and b,b' there always exists m such that a>=b-m and a'<=b'+m, like m>=max(|a-b|,|a'-b'|). Am I wrong? I think I do not understand this assumption...**

Yes, Assumption 1 is trivial. The key element to note from Assumption 1 is that there is some $m$ (that could be very large) where this inequality holds. Now, the question is how does this constant $m$ affect the analysis? Larger $m$ results into requiring a larger input dimension $n$ for Theorem 1 to hold. To see this, refer to the last equation of the proof of Theorem 1. Note that the second term of the right hand side of the inequality is a function of $m$. This shows that the larger the constant $m$ for the Assumption 1 to hold, the larger the input dimension for the inequality to be positive and thereafter for Theorem 1 to hold.

• **How simple is it to extend those theoretical results for NNs to the case of CNNs?**

The results hold for generic linear operators $\mathbf{A}$ and $\mathbf{a}_2$. For convolutional layers, the kernel can indeed be represented with a structured topelitz/circulant matrix fitting into our analysis. However, we find a trick to avoid the inefficient construction of such massive matrices. Note that to compute our bounds, one only needs to compute Eq (3) efficiently where $\mathbf{A}_1$ is convolutional kernel. For $\mathbf{A}_1\mathbf{x}$, this can be computed by simply performing convolution to the input center $\mathbf{x}$ without constructing $\mathbf{A}_1$ (which is a forward pass through the layer). The tricky part comes upon computing the term $|\mathbf{a}_2^\top \mathbf{M} \mathbf{A}_1| \mathbf{1}_n$. As one need to compute $|\mathbf{a}_2^\top \mathbf{M} \mathbf{A}_1|$ first where $\mathbf{M}$ is a diagonal matrix without constructing $\mathbf{A}_1$. The way we go about this is by simply taking a backward pass through the layer that computes $\mathbf{a}_2^\top \mathbf{M} \mathbf{A}_1$ for some input center $\mathbf{x}$ as a function of $\mathbf{x}$. That is we perform a forward pass for some input center $\mathbf{x}$ and then compute the gradients as a function of $\mathbf{x}$ resulting in having access to $\mathbf{a}_2^\top \mathbf{M} \mathbf{A}_1$ and thereafter computing $|\mathbf{a}_2^\top \mathbf{M} \mathbf{A}_1| \mathbf{1}_n$ becomes efficient. Note that this has been employed in the experiments as the networks used did have convolutional layers.

## G.3 REVIEWER 3

We thank R3 for their time reviewing the paper. Follows our response.

• **However, these results shown in Figs 6 and 7 appear to be somewhat inconsistent with those reported in the paper of IBP [Gowal et al., 2018)].**

We would like to raise to the attention of R3 that the metric used here is PGD robustness and not PGD accuracy. While they are to some extent related, the PGD robustness is the percentage of the testing samples where the network prediction was not altered under PGD attacks from the prediction of the network on the original samples regardless if the prediction is correct. Since the pretrained models of IBP were not made available, we trained models with IBP by running the code provided by the authors on their github page (with the same exact setup). Moreover, for a fair comparison and stronger baselines, all IBP trained models were trained over different learning schedules. The best models with the highest accuracy-robustness pair trade-off for IBP are reported.

• **I am not sure if it really makes sense to discuss which method is better in such ranges of failed defense.**

We believe that so long the model accuracy on the noise free-samples is still preserved (captured on the x-axis of Figures 5 and 6) with an increase in robustness, this is good evidence for improvement. For instance, for CIFAR10 on the small architecture (Figure 6), our models have improved the average robustness over all $\epsilon_{\text{test}}$ (captured by y-axis) while preserving similar accuracy to the nominal model. This is unlike IBP, where despite that the robustness improved $30\%$ the accuracy dropped by $20\%$ compared to the nominal model.

• **Sources for differences to Gowal et. al.**

Indeed, as correctly pointed by R3, the reported robustness is averaged over all tested $\epsilon$. However, note that we report the robustness results for individual $\epsilon_{\text{test}}$ in the supplementary material (Figures 10-17). The range of $\epsilon_{\text{test}}$ for MNIST is identical to that of Gowal et. al.; however, the range for $\epsilon_{\text{test}}$ is indeed larger for CIFAR10 experiments.

- **Comment on the metric PGD Robustness and not PGD accuracy.**

The focus of our work is to show an efficient approach to estimate the output bounds of a network. Since the bounds are only 'true' in expectation they can not be used for certification in the absence of concentration results to our bounds. To that extent, verified accuracy (accuracy by running exact solvers) and its lower and upper bound estimates, certified accuracy and PGD accuracy respectively, do not make much sense in our framework. Therefore, we limit our experiments to showcasing that such simple estimates of the bounds can be utilized to efficiently regularize networks to improve their robustness.

