# OpenReview forum: "Expected Tight Bounds for Robust Deep Neural Network Training"
_ICLR.cc/2020/Conference — Reject_

### Official Review · AnonReviewer3 · 2019-10-22
**Official Blind Review #3**

**Rating:** 3

**Review:**

I understand that the proposed bound is derived based on a few assumptions, which do not necessarily hold true in real cases, and thus it is only an approximate bound, meaning that the bound is not theoretically guaranteed to be a ‘superset’ of the true bound. In fact, in both Fig.1 (dealing with ideal conditions) and Table 1 (dealing with a real task (MNIST)), \Gamma, which is an indicator that should be 1 if the proposed bound is the superset, is not always 1.

Therefore, it is fair to say that the practical value of the proposed approach depends on its performance on real tasks. This is supposed to be judged mainly from Fig.6 (MNIST) and Fig.7 (CIFAR-10). My first impression is that it is hard to see the expected accuracy-robustness trade-offs from them and the results appear rather random. That said, there are several cases for which the proposed method show better accuracy and robustness than IBP, as is claimed in the paper.

However, these results shown in Figs 6 and 7 appear to be somewhat inconsistent with those reported in the paper of IBP [Gowal et al., 2018)]. They are summarized in Table 4 of [Gowal et al., 2018)], where IBP achieves PCD accuracy 97.9, 96.1, 93.9, and 89.7% on MNIST for \epsilon_{test}=0.1, 0.2, 0.3, and 0.4, respectively, and achieves 55 and 35% on CIFAR-10 for \epsilon_{test}=2/225 and 8/225, respectively. On the other hand, in Figs 6 and 7 of this paper, the PCD accuracies of IBP and the proposed method distribute in the range of 30-70% for MNIST (Fig.6) and 5-20% for CIFAR-10 (Fig.7). Considering that the accuracy under no attack is greater than 98% on MNIST and 50% on CIFAR-10, we should consider that the defense practically fails in these ranges. I am not sure if it really makes sense to discuss which method is better in such ranges of failed defense.

This difference from [Gowal et al., 2018)] seems to come from i) the employment of the average PGD robustness over all \epsilon_{test}’s for Figs. 6 and 7, as in the statement: ”To compare training methods, we compute the average PGD robustness over all εtest and the test accuracy, and report them in a 2D scatter plot“; and ii) the range of \epsilon_{test} is wider than [Gowal et al., 2018)]. I think that the evaluation in [Gowal et al., 2018)] is conducted in the range that the defense is regarded to be successful, and thus is more appropriate.


**Experience Assessment:**

I have published in this field for several years.

**Review Assessment: Checking Correctness Of Derivations And Theory:**

I assessed the sensibility of the derivations and theory.

**Review Assessment: Checking Correctness Of Experiments:**

I carefully checked the experiments.

**Review Assessment: Thoroughness In Paper Reading:**

I read the paper at least twice and used my best judgement in assessing the paper.

---

> ### Author Response · Authors · 2019-11-14
> **Response to R3**
>
> We thank R3 for their time reviewing the paper. Follows our response.
>
>
> $\bullet$ However, these results shown in Figs 6 and 7 appear to be somewhat inconsistent with those reported in the paper of IBP [Gowal et al., 2018)].
>
> We would like to raise to the attention of R3 that the metric used here is PGD robustness and not PGD accuracy. While they are to some extent related, the PGD robustness is the percentage of the testing samples where the network prediction was not altered under PGD attacks from the prediction of the network on the original samples regardless if the prediction is correct. Since the pretrained models of IBP were not made available, we trained models with IBP by running the code provided by the authors on their github page (with the same exact setup). Moreover, for a fair comparison and stronger baselines, all IBP trained models were trained over different learning schedules. The best models with the highest accuracy-robustness pair trade-off for IBP are reported.
>
>
> $\bullet$ I am not sure if it really makes sense to discuss which method is better in such ranges of failed defense.
>
> We believe that so long the model accuracy on the noise free-samples is still preserved (captured on the x-axis of Figures 5 and 6) with an increase in robustness, this is good evidence for improvement. For instance, for CIFAR10 on the small architecture (Figure 6), our models have improved the average robustness over all $\epsilon_{\text{test}}$ (captured by y-axis) while preserving similar accuracy to the nominal model. This is unlike IBP, where despite that the robustness improved $30\%$ the accuracy dropped by $20\%$ compared to the nominal model.
>
>
> $\bullet$ Sources for differences to Gowal et. al.
>
> Indeed, as correctly pointed by R3, the reported robustness is averaged over all tested $\epsilon$. However, note that we report the robustness results for individual $\epsilon_{\text{test}}$ in the supplementary material (Figures 10-17). The range of $\epsilon_{\text{test}}$ for MNIST is identical to that of Gowal et. al.; however, the range for $\epsilon_{\text{test}}$ is indeed larger for CIFAR10 experiments.
>
> $\bullet$ Comment on the metric PGD Robustness and not PGD accuracy.
>
> The focus of our work is to show an efficient approach to estimate the output bounds of a network. Since the bounds are only 'true' in expectation they can not be used for certification in the absence of concentration results to our bounds. To that extent, verified accuracy (accuracy by running exact solvers) and its lower and upper bound estimates, certified accuracy and PGD accuracy respectively, do not make much sense in our framework. Therefore, we limit our experiments to showcasing that such simple estimates of the bounds can be utilized to efficiently regularize networks to improve their robustness.

---

### Official Review · AnonReviewer1 · 2019-10-23
**Official Blind Review #1**

**Rating:** 3

**Review:**

This work proposes some expected bounds in order to improve the robustness-accuracy trade-off on standard CNNs. The bounds numerically improve (Gowal et al), thus this method works numerically well.

My main concern is about the fact that several assumptions seem relatively constraining or are not quantitatively justified. Furthermore, I'm wondering if such bounds could be applied to Neural Tangent Kernel works. This could be useful and it would seem to correspond better to the setting of this work.

Pros:
- The performances are quite good.
- In expectation for random NNs, the proposed bounds are tighter.
- The beginning of Section 4 conducts an interesting numerical study that tries to validate those bounds.

Cons:
- Some assumptions seem pretty unrealistic to me. For instance, the fact that the neural network should have Gaussian i.i.d weights: it is thus surprising that this technique works in real-life settings. As far as I understood, this assumption is not done in (Gowal et al). Did the authors check that their CNNs had Gaussian weights? I'm not convinced by the paragraph at the end of section 3.2, yet a simple histogram to validate this claim would convince me.
- A major difference with (Gowal et al) is that: here, the bounds are in expectation whereas the bound in (Gowal et al) are deterministic. In this paper, there are some approximation assumptions (e.g., large input, large number of hidden layers) without non-asymptotic arguments. For instance, I was sort of expecting some concentration inequalities that would allow to quantify how large should be the input dimension $n$ or the width $k$.
- Most of the proofs are long and complicated, and several equations of 7-8 lines could be summarized with up to 2-3 lines maximum. I had a hard time to understand the proof of Theorem 1, which is simply some algebra. This could be improved.
- I'm curious of the imagenet performance: couldn't this technique be easily applied to AlexNet, which is nowadays simple to manipulate? Is there a technical issue to do so?
- Is the assumption 1 really an assumption? given a,a' and b,b' there always exists m such that a>=b-m and a'<=b'+m, like m>=max(|a-b|,|a'-b'|). Am I wrong? I think I do not understand this assumption...
- How simple is it to extend those theoretical results for NNs to the case of CNNs?

**Experience Assessment:**

I have read many papers in this area.

**Review Assessment: Checking Correctness Of Derivations And Theory:**

I carefully checked the derivations and theory.

**Review Assessment: Checking Correctness Of Experiments:**

I carefully checked the experiments.

**Review Assessment: Thoroughness In Paper Reading:**

I read the paper thoroughly.

---

> ### Author Response · Authors · 2019-11-14
> **Response (1/2) to R1**
>
> We thank R1 for the time spent in reviewing our paper. Follows our response to R1's concerns.
>
> $\bullet$ Some assumptions seem pretty unrealistic to me. For instance, the fact that the neural network should have Gaussian i.i.d weights: it is thus ...
>
> As correctly pointed out by R1, Gowal et al do not make such an assumption. They provide loose 'true' bounds for deep architectures as a function of network parameters by simply computing worst case bound per layer. As suggested by R1, we have conducted an experiment where we trained Medium CNN with identical structure to previous experiments on MNIST, CIFAR10 and on CIFAR100 with and without $\ell_2$ regularization. We then plot the histogram of the weights of the 6 layers in the network (3 convolutional and 3 fully connected). We report the results in Section E of the appendix in the revised version. For the MNIST experiments, even without $\ell_2$ regularization, the weights seem to follow a bell shaped. With $\ell_2$ regularization, the histogram tend to be much pointy. Of course, the regularization parameter used in training interpolates between the two histograms, the larger the regularization the more 'pointy' the histogram will tend to be. A similar observation can be noted for CIFAR10 and CIFAR100. We want to emphasize that while this is not a proof that the Gaussian assumption holds in practice, we believe that at least the distribution of the weights does not seem to be far from Gaussian and thereafter sheds light on why perhaps such an assumption may not be utterly unreasonable on real networks.
>
>
> $\bullet$ A major difference with (Gowal et al) is that: here, the bounds are in expectation whereas the bound in (Gowal et al) are deterministic. In this paper, there are some approximation assumptions ...
>
> This is true. Unfortunately, as far as the analysis goes, we do not yet have concentration results. One direction of interest is to study the asymptotic behaviour (in terms of $n$ and $k$) of the variance of $\mathbf{L}_{\text{M}}$ and $\mathbf{U}_{\text{M}}$ over the distribution of network parameters. So far we have some little progress in that direction as the analysis becomes untraceable.
>
>
> $\bullet$ Most of the proofs are long and complicated, and several equations of 7-8 lines could be summarized with up to 2-3 lines maximum. I had a hard time to understand the proof of Theorem 1, which is simply some algebra. This could be improved.
>
> The equations were, for most part purposely, left to span multiple lines as we wanted to elaborate on each step of the analysis.
>
>
> $\bullet$ I'm curious of the imagenet performance: couldn't this technique be easily applied to AlexNet, which is nowadays simple to manipulate? Is there a technical issue to do so?
>
> While we were interested for the most part in the analysis, we do believe that the paper can benefit from the ImageNet experiments vastly. They were left out for time/computation capacity related reasons. We will try to address this within our capacity.
>
>
>
> $\bullet$ Is the assumption 1 really an assumption? given a,a' and b,b' there always exists m such that a>=b-m and a'<=b'+m, like m>=max(|a-b|,|a'-b'|). Am I wrong? I think I do not understand this assumption...
>
> Yes, Assumption 1 is trivial. The key element to note from Assumption 1 is that there is some $m$ (that could be very large) where this inequality holds. Now, the question is how does this constant $m$ affect the analysis? Larger $m$ results into requiring a larger input dimension $n$ for Theorem 1 to hold. To see this, refer to the last equation of the proof of Theorem 1. Note that the second term of the right hand side of the inequality is a function of $m$. This shows that the larger the constant $m$ for the Assumption 1 to hold, the larger the input dimension for the inequality to be positive and thereafter for Theorem 1 to hold.

---

> > ### Author Response · Authors · 2019-11-14
> > **Response (2/2) to R1**
> >
> > $\bullet$ How simple is it to extend those theoretical results for NNs to the case of CNNs?
> >
> > The results hold for generic linear operators $\mathbf{A}$ and $\mathbf{a}_2$. For convolutional layers, the kernel can indeed be represented with a structured topelitz/circulant matrix fitting into our analysis. However, we find a trick to avoid the inefficient construction of such massive matrices. Note that to compute our bounds, one only needs to compute Eq (3) efficiently where $\mathbf{A}_1$ is convolutional kernel. For $\mathbf{A}_1\mathbf{x}$, this can be computed by simply performing convolution to the input center $\mathbf{x}$ without constructing $\mathbf{A}_1$ (which is a forward pass through the layer). The tricky part comes upon computing the term $|\mathbf{a}_2^\top \mathbf{M} \mathbf{A}_1|\mathbf{1}_n$. As one need to compute $|\mathbf{a}_2^\top \mathbf{M} \mathbf{A}_1|$ first where $\mathbf{M}$ is a diagonal matrix without constructing $\mathbf{A}_1$. The way we go about this is by simply taking a backward pass through the layer that computes $\mathbf{a}_2^\top \mathbf{M} \mathbf{A}_1$ for some input center $\mathbf{x}$ as a function of $\mathbf{x}$. That is we perform a forward pass for some input center $\mathbf{x}$ and then compute the gradients as a function of $\mathbf{x}$ resulting in having access to $\mathbf{a}_2^\top \mathbf{M} \mathbf{A}_1$ and thereafter computing $|\mathbf{a}_2^\top \mathbf{M} \mathbf{A}_1|\mathbf{1}_n$ becomes efficient. Note that this has been employed in the experiments as the networks used did have convolutional layers.

---

> > > ### Comment · AnonReviewer1 · 2019-11-14
> > > **Thanks for your rebuttal**
> > >
> > > 1/ I understand but then I think the theory of the paper is not as strong as it sounds. A stronger argument w.r.t. the modelization would make me encline to accept this statement.
> > >
> > > 2/ Thanks for your answer.
> > >
> > > 3/ Right. I spent a significant amount of time to validate them..
> > >
> > > 4/ I still don't understand the issue with ImageNet, for instance, an AlexNet is not significantly more difficult to deploy. That would leverage my concerns w.r.t. 1/ as this paper would definitely fall in the category of empirical papers.
> > >
> > > 5/ Right I think I understood this. Then maybe, if this always true, maybe it could not be called "assumption".. simply replacing it with a discussion on the value of $m$..
> > >
> > > 6/ Thanks, I see: thus no specific adaptation.(I was slightly unsure but I find your arguments convincing, thanks)

---

### Official Review · AnonReviewer2 · 2019-10-24
**Official Blind Review #2**

**Rating:** 3

**Review:**

Disclaimer: I have already reviewed this paper for another conference. I re-read it to assess the modifications made by the authors but I am aware of the comments that they received in the previous round of reviews. I'm also re-raising some concerns that I made in a previous review that the authors didn't address.

Summary
Interval Bound Propagation (IBP) is a fast method to propagate bounds through the activation of a Neural Network. It is however quite loose. The authors of this paper propose a different way of propagating the bounds, which is not a rigorous bound computation method, but for which they show that, with some strong assumptions on the distribution of the weights, the expectation of the generated results are valid bounds that are tighter than the ones generated by IBP.

The paper explains clearly the "how" of how these bounds are achieved. Equation 3 and the paragraph before it are good. (Essentially, as long as a ReLU is not in a "blocking" state where all its input are negative, treat it as an identity, and do a forward pass of the center of the input region).
The logical argument explaining the "why" these are valid is harder to understand and could be clarified. My understanding so far is:
- All the analysis is dependent on all weights being drawn from iid gaussian of zero mean.
- In addition to that, there is an assumption (Assumption 1) between the relation of bounds of network with random weight and some quantity (L_approx, U_approx) with random uniform inputs x.
The proof is done that the expectation of the bound proposed is "correct" with regards to (L_approx, U_approx) but then, the relation to the true bounds is given by the Assumption?
So the relation to the true bounds is only given by the Assumption?

Regarding the experiments:
* The caption of Figure 1 is misleading. It says that it shows that the proposed bounds are a super set of the interval bounds, while they are actually not (ratio is strictly smaller than 1). You can argue that they are close, but now that they are a superset when they aren't!
* The reporting of Figure 5 and 6 is weird because according to the text, each datapoint seems to be the average robustness of networks trained with different hyperparameters, so it's hard to interpret.
* I appreciate the effort of the author to include experiments involving a MIP solver returning the true bounds. This is very helpful in building confidence that the bounds generated are correct. How is the real network trained? Is this based on a robustly trained network or is it just standard training? Is 99% the nominal accuracy?

In general, I think that the paper would be better if there was more discussion of the failure modes of the method. There are easy to identify failure cases where the proposed bound is incorrect or loose. My opinion is that the paper would be stronger if it acknowledged them but then made the points that the bounds proposed are still good most of the time (which their experiment show) and useful (for example for training where the exact correctness of the bounds may not be the most important), rather than ignoring them and pretending that the bounds are perfect.
[From previous review]
Simple example of a network where the results would be quite loose, while IBP is tight:
Two layer NN, x = 0 \in R^n, eps = 1, W1 = 1000 * Identity, b1 = -999 * 1_n, a2 = -10 1_{1,n} , b2 = 0
u1 would be all positive, so the matrix M would simply be the identity.
L_M = a2^T M b1 - eps | a2^T M A1| vec(1) = (-10)*(-999)*n - 10 * (1000) * n = -19990 * n
U_M = a2^T M b1 + eps | a2^T M A1| vec(1) = (-10)*(-999)*n + 10 * (1000) * n = n
The actual bound (which also correspond in that case to what would be computed by IBP) is
L_gt = -10*n
u_gt = 0

Comments:
* "We prove to be true bounds in expectation" -> This is a bad formulation that should be rephrased. The expectation of the proposed bound is correct with regards to the expectation of the true bounds, but "True in expectation" is a bad formulation that doesn't reflect what is happening. Maybe True in expectation would mean that E_{A_1, a_2}(indicator(L_true > L_M)) -> 1 as n grows

* The assumption about gaussian weights in A1 and a2 seems strong but is at least partially motivated at the end of 3.2 (although not all networks are trained with l2 regularizer) What about the iid assumption? In case of a CNN for example, the weights are shared, so definetely won't fit this framework?

* I think that the paper would benefit from having some discussion of other methods that can derive "bounds" which may not actually be bounds. The works by Stefan Webb (A Statistical Approach to Assessing Neural Network Robustness, ICLR 2019) or Tsui Wei Weng (Evaluating the Robustness of Neural Networks: An Extreme Value Theory Approach, ICLR 2018) being others of the top of my head . While this paper clearly propose to do things very differently, I think the discussion would have been very valuable.

* While reviewing the paper, I also spotted some strong similarities between the methods proposed (particularly subsection 3.3) and the Fastlin method. The computation mechanism of Fastlin (propagate recursively  from the end through the linear layers and through diagonal matrices that replace the ReLU activation function) is exactly the same as the proposed one (Fastlin goes through the notational burden of handling the bias). The only difference is in the way the coefficients of the diagonal matrix are computed. (here, it's 0 if the Relu is blocking, and 1 otherwise. Fastlin has 0 if blocking, 1 if passing but a coefficient in between (u/u-l) if the reLU is ambiguous).
If that connection is correct, then the benefits of the proposed methods are limited: neither Fastlin nor IBP dominate each other, and the computational costs for Fastlin is higher, due to having to propagate the full linear maps here noted G from the end to the input to obtain the bounds, as the authors seem to have also realised ("it is expensive to compute our bounds using the procedure in Section 3.3, so instead, we obtain matrices Mi using the easy-to-compute IBP upper bounds.")

Opinion:
I am bothered by the framing of the results that the authors employ, as their theoretical results are, to the best of my understanding, dependent on strong assumptions. Some of the experimental results are also over-exaggerated (caption saying that things are superset while the experimental results show something different), and the relation and contextualization with regards to existing literature is lacking ( both against other works with similar aims but different methods, and with works with similar method but a slightly different aim)

Typos:
Page 7, "are are much"
Page 8,"When kappa=0", no need for uppercase
The references should ideally be cleaned and put in a consistent format. In the text of the paper, Some citation have first name + last name of one author, some have two out of all the authors, some have the et al. format.... It's all over the place.
In the References section, some authors list are shuffled (at least the "Scaling provable adversarial" paper), some are missing the conference where paper where cited and just cite the arxiv version

**Experience Assessment:**

I have published in this field for several years.

**Review Assessment: Checking Correctness Of Derivations And Theory:**

I assessed the sensibility of the derivations and theory.

**Review Assessment: Checking Correctness Of Experiments:**

I carefully checked the experiments.

**Review Assessment: Thoroughness In Paper Reading:**

I read the paper thoroughly.

---

> ### Author Response · Authors · 2019-11-14
> **Response (1/2) to R2**
>
> We thank R2 for their very thorough review of our paper. Regarding the disclaimer, we want to clarify to other reviewers too that we have indeed tried to address all R2's previous concerns in this submission. We have conducted some extra experiments from the previous submission; we have also done some serious changes to the structure and presentation of the paper as previously suggested by R2. In what follows we try addressing R2's current concerns.
>
>
> $\bullet$ The logical argument explaining the "why" these are valid is harder to understand and could be clarified. My understanding so far ...
>
> The overall description of R2 to the logical argument is correct. The proof is based on showing that our bounds are 'supersets' to some approximate bounds $\mathbf{L}_{\text{approx}}$ and $\mathbf{U}_{\text{approx}}$. The approximate bounds are indeed related to the unknown true bounds through Assumption 1. Note that Assumption 1 states that there exists some $m$ where the Assumption inequality holds. As we pointed out to R1, the larger the constant $m$, the larger $n$ has to be for Theorem 1 to hold. Please refer to the last inequality in the proof of Theorem 1.
>
>
> $\bullet$ The caption of Figure 1 is misleading. It says that it shows that the proposed ...
>
> True. This was a mistake of phrasing on our part. We have addressed this in the revised version.
>
>
> $\bullet$ The reporting of Figure 5 and 6 is weird because according to the text, each datapoint seems to be the average robustness of ...
>
> We have addressed this in the revised version. The reported models had only their robustness averaged over multiple $\epsilon_{\text{test}}$. Please note that the robustness results over each individual $\epsilon_{\text{test}}$ were left for the appendix. The best models from the grid search over the hyperparameters are reported for both our bound training and IBP training for fair comparison.
>
>
> $\bullet$ I appreciate the effort of the author to include experiments involving a MIP solver returning the true bounds. This is very helpful in building confidence that the bounds generated are correct. How is the real network trained? Is this based on a robustly trained network or is it just standard training? Is 99$\%$ the nominal accuracy?
>
> Yes, $99\%$ is the nominal accuracy of the trained models. The network was trained normally without any special regularization.
>
>
> $\bullet$ In general, I think that the paper would be better if there was more discussion of the failure modes of the method. There are easy to identify failure cases where the proposed bound is incorrect or loose. My opinion is that the paper would be stronger if it acknowledged them ...
>
> We agree with R2. We have addressed this in the revised version. In particular, we reflected this in the paragraph just before section 3.3. We have added in the appendix the example of the failure case pointed out by R2. Moreover, as suggested by R1, we have investigated, at least empirically, the histogram of the weights of a normally trained network on 3 real dataset in appendix F. In what follows, we re-state our response to R1 in regards to the histogram results of the appendix to R2 for completion. "For the MNIST experiments, even without $\ell_2$ regularization, the weights seem to follow a bell shaped. With $\ell_2$ regularization, the histogram tend to be much pointy. Of course, the regularization parameter used in training interpolates between the two histograms, the larger the regularization the more 'pointy' the histogram will tend to be. A similar observation can be noted for CIFAR10 and CIFAR100. We want to emphasize that while this is not a proof that the Gaussian assumption holds in practice, we believe that at least the distribution of the weights does not seem to be far from Gaussian and thereafter sheds light on why perhaps such an assumption may not be utterly unreasonable on real networks."
>
>
> $\bullet$ The assumption about gaussian weights in A1 and a2 seems strong but is at least partially motivated at the end of 3.2 (although not all networks are ...
>
> R2 is correct. The i.i.d assumption over all elements of the weights matrix $\mathbf{A}$ breaks in the presence of topelitz/circulant structure convolutional layers have. This, unfortunately, is as far as the current theory can go. We find though that this have little impact on practically upon training the networks.

---

> > ### Author Response · Authors · 2019-11-14
> > **Response (2/2) to R2**
> >
> >
> > $\bullet$ I think that the paper would benefit from having some discussion of other methods that can derive "bounds" which may not actually be bounds. The works by Stefan Webb (A Statistical ...
> >
> >
> > Indeed, the two referenced works are related to our direction since both aim at either probabilistically carrying out the verification or by estimating the lower bound of the minimum perturbation. We elaborated on this in the revised version and added a paragraph in the introduction.
> >
> > $\bullet$ While reviewing the paper, I also spotted some strong similarities between the methods proposed (particularly subsection 3.3) and the Fastlin method. The computation mechanism of Fastlin (propagate recursively  from the end through the linear layers and through diagonal matrices that replace the ReLU activation function) is exactly the same as the proposed one (Fastlin goes through the ...
> >
> > The description of R1 for the intuition of the construction of the diagonal matrix $\mathbf{M}$ is correct. Indeed, there are similarities between both our work and Fastlin. However, we do not believe that the works are limited. Both works still provide an excellent trade-off between efficiency and computational complexity, even when over estimating the intermediate upper bounds for the intermediate $\mathbf{M}$, the approximation for the superset bounds are still effective.
> >
> >
> > $\bullet$ Minor comments, typos and references.
> > We have addressed the issues in the revised version. We have also tried to address other issues and writing style highlighted by R2.

---

### Author Response · Authors · 2019-09-30
**Link to code**

While it is very unlikely unless it was purposely looked for, the affiliation of the authors could potentially be exposed in the original link to the code provided.

To this end, we eliminate this possibility where we have disabled the link sharing. We provide the new link to code here

https://drive.google.com/file/d/1ufoipDPOqwCD32CQVgGr_2l39-4pnRTS/view?usp=sharing

---

### Decision · Program_Chairs · 2019-12-19

**Decision:**

Reject

**Comment:**

The authors propose a new technique for training networks to be robust to adversarial perturbations. They do this by computing bounds on the impact of the worst case adversarial attack, but that only hold under strong assumptions on the distribution of the network weights. While these bounds are not rigorous, the authors show that they can produce networks that improve the robustness-accuracy tradeoff on image classification tasks.

While the idea proposed by the authors is interesting, the reviewers had several concerns about this paper:
1) The assumptions required for the bounds to hold are unrealistic and unlikely to hold in practice, especially for convolutional neural networks.
2) The comparisons are not presented in a fair manner that allow the reader to interpret the difference between the nature of certificates computed by the authors and those computed in prior work.
3) The empirical gains are not substantial if one normalizes for the non-rigorous nature of the certificates computed (given that they only hold under hard-to-justify assumptions).

The rebuttal phase clarified some issues in the paper, but the fundamental flaws with the approach remain unaddressed. Thus, I recommend rejection and suggest that the authors revisit the assumptions and develop more convincing arguments and/or experiments justifying them for practical deep learning scenarios.